# Structural basis of gap-filling DNA synthesis in the nucleosome by DNA Polymerase β

Tyler M. Weaver [1,2,6], Benjamin J. Ryan[1,6], Spencer H. Thompson [1], Adil S. Hussen[1], Jonah J. Spencer[1], Zhen Xu [3], Nicholas J. Schnicker [3,4] & Bret D. Freudenthal [1,2,5] ✉

Single-strand breaks (SSBs) are one of the most prevalent forms of DNA damage found in the chromatinized genome and are repaired by single-strand break repair (SSBR) or base excision repair (BER). DNA polymerase beta (Pol β) is the primary enzyme responsible for processing the 1-nt gap intermediate in chromatin during SSBR and BER. To date, the mechanism used by Pol β to process a 1-nt gap in the context of chromatin remains poorly understood. Here, we use biochemical assays and cryogenic electron microscopy (cryo-EM) to determine the kinetic and structural basis of gap-filling DNA synthesis in the nucleosome by Pol β. This work establishes that Pol β uses a global DNA sculpting mechanism for processing 1-nt gaps in the nucleosome during SSBR and BER, providing fundamental insight into DNA repair in chromatin.

Genomic DNA is packaged into chromatin through a fundamental repeating unit known as the nucleosome core particle (NCP). The NCP is an octameric protein-nucleic acid complex composed of two copies of histone H2A, H2B, H3, and H4 wrapped by ~147 bp of nucleosomal DNA[1], which serves as a regulatory barrier for accessing genomic DNA. Chromatinized genomic DNA is continuously exposed to a variety of endogenous and exogenous stressors that generate DNA damage. This DNA damage occurs ubiquitously throughout chromatinized and non-chromatinized regions of genomic DNA[2–12], and must be efficiently repaired to maintain genome stability.

Single-strand breaks (SSBs) are one of the most prevalent forms of genomic DNA damage, occurring tens of thousands of times per cell per day[13]. If left unrepaired, SSBs can result in mutagenesis, genome instability, and/or cell death[14–17]. To protect the genome, the cell possesses robust single-strand break repair (SSBR) pathways to identify, process, and repair SSBs that arise from various endogenous and exogenous sources[16,17]. One common mechanism for the generation of SSBs is through decomposition of the sugar-phosphate backbone via oxidation[18], which requires subsequent DNA end processing prior to being repaired directly by the SSBR pathway[17]. SSBs can also form

indirectly via the base excision repair (BER) pathway, which is responsible for repairing oxidative and alkylative base damage[19,20]. These indirect SSBs arise through the enzymatic activity of bifunctional DNA glycosylases or the combined action of a monofunctional DNA glycosylase and AP-endonuclease I (APE1) that creates a 3'-hydroxyl and 5'-deoxyribosephosphate (dRP) nick termini. This substrate is subsequently processed by the AP-lyase activity of DNA polymerase beta (Pol β) creating a 1-nt gap[21,22].

Though the source of the initial SSB and the upstream end processing steps differ between direct SSBR and BER, both repair pathways converge on a common DNA intermediate containing a 1-nt gap that is primarily processed by Pol β[16,23–26]. During BER and SSBR, Pol β binds the 1-nt gap repair intermediate through interactions with the 5'-phosphate and the primer terminal 3'-OH, which is mediated by the lyase (i.e., 8-kDa domain) and polymerase domain (i.e., 31-kDa domain), respectively[27,28]. During subsequent gap-filling DNA synthesis, Pol β binds the correct nucleotide triphosphate and divalent cation metals (e.g., $Mg^{2+}$) in the polymerase active site and rapidly catalyzes nucleotide insertion to fill the 1-nt gap[29]. Following Pol β nucleotide insertion, the remnant 3'-nick repair intermediate is sealed

[1]Department of Biochemistry and Molecular Biology, University of Kansas Medical Center, Kansas City, KS 66160, USA. [2]Department of Cancer Biology, University of Kansas Medical Center, Kansas City, KS 66160, USA. [3]Protein and Crystallography Facility, University of Iowa Carver College of Medicine, Iowa City, IA 52242, USA. [4]Department of Molecular Physiology and Biophysics, University of Iowa Carver College of Medicine, Iowa City, IA 52242, USA. [5]University of Kansas Cancer Center, Kansas City, KS 66160, USA. [6]These authors contributed equally: Tyler M. Weaver, Benjamin J. Ryan. ✉e-mail: bfreudenthal@kumc.edu

by DNA Ligase I or DNA Ligase IIIα successfully completing repair of the SSB[17,19].

The mechanism used by Pol β to engage 1-nt gaps and catalyze gap-filling DNA synthesis in the context of non-chromatinized DNA has been extensively characterized[20,29,30], however, the mechanism used in the context of chromatin remains poorly understood. Initial evidence of Pol β function in chromatin was described almost 50 years ago, when Pol β was found to associate with nucleosomes from micrococcal nuclease digested cellular chromatin[31]. Contemporary in vitro studies with recombinant nucleosomes containing 1-nt gaps identified that Pol β can perform gap-filling DNA synthesis in the nucleosome, though with a reduced efficiency compared to non-nucleosomal DNA[32–41]. In contrast, the 5′-dRP lyase activity of Pol β has a similar catalytic efficiency in both nucleosomal and non-nucleosomal DNA[34]. Despite these prior biochemical studies, the structural mechanism used by Pol β to engage 1-nt gaps and perform gap-filling DNA synthesis in the nucleosome remains unknown. Here, we use a combination of biochemical assays and cryogenic electron microscopy (cryo-EM) to determine the kinetic and structural basis of gap-filling DNA synthesis in the nucleosome by Pol β, providing fundamental insight into the processing of 1-nt gaps in chromatin during SSBR and BER.

## Results

To investigate the ability of Pol β to perform gap-filling DNA synthesis in the nucleosome, we generated eight NCPs with 1-nt gaps (Gap-NCPs) at varying positions within a 147 bp Widom 601 strong positioning sequence (Fig. 1a, b and Supplementary Fig. 1a)[42]. Five of the NCPs contain a solvent-exposed 1-nt gap in a unique translational position (i.e., position relative to the nucleosome dyad) at SHL−5.5, SHL−4.5, SHL−3.5, SHL−2.5, and SHL−1.5. The three additional NCPs have a 1-nt gap at SHL−4.5 that differ in rotational orientation relative to the histone octamer (i.e., solvent-exposed or occluded), termed SHL−4.5+2, SHL−4.5+3, and SHL−4.5+4. Importantly, the translational position and rotational orientation of these 1-nt gaps were defined based on previously determined cryo-EM structures of NCPs with the same nucleosomal DNA sequence[43,44]. These substrates were then used to systematically determine the ability of Pol β to bind 1-nt gaps and catalyze gap-filling DNA synthesis in the nucleosome.

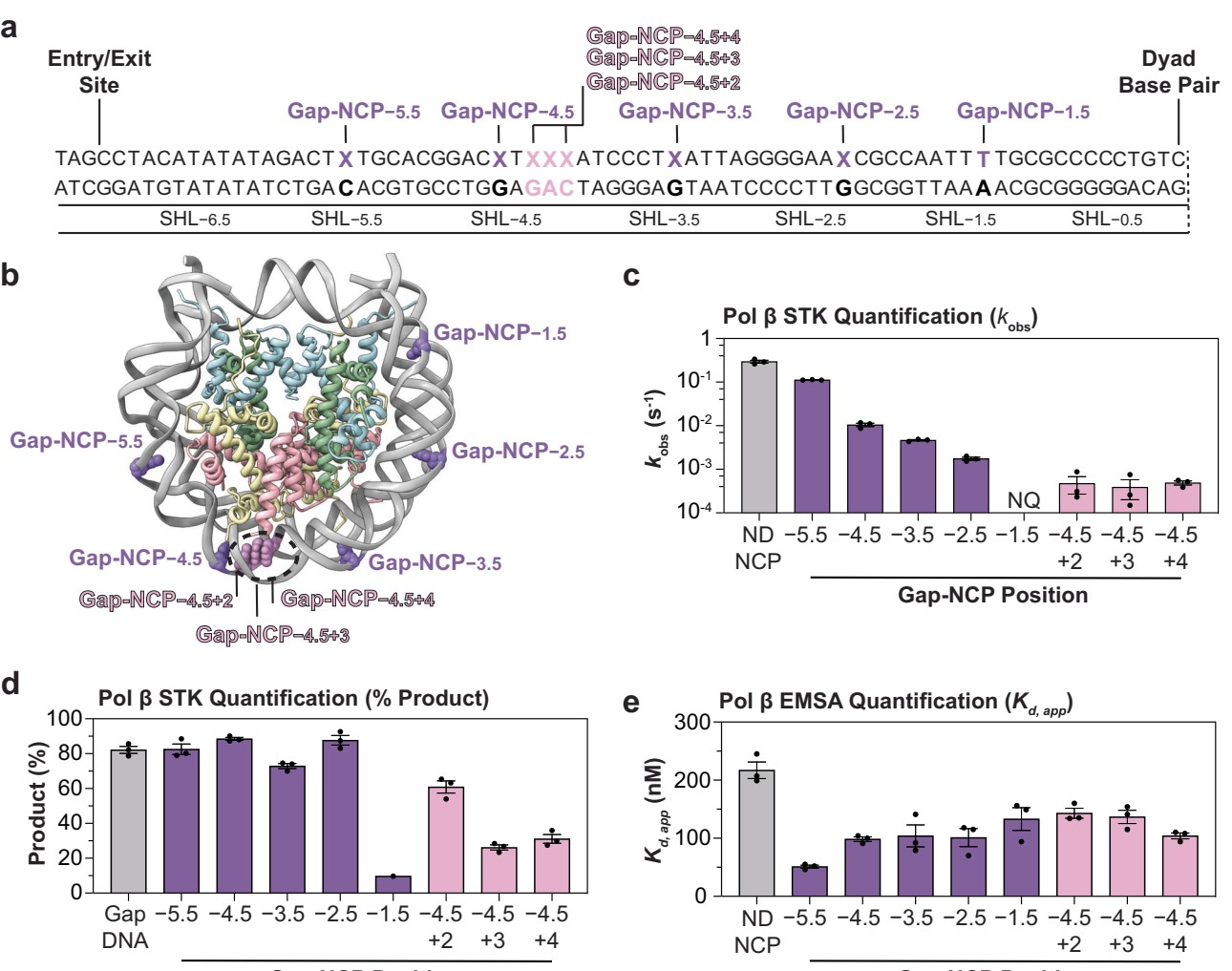

**Fig. 1 | Biochemical analysis of Pol β nucleosome binding and gap-filling DNA synthesis. a** Diagram of the 601 strong positioning sequence highlighting the position of each individual 1-nt gap (denoted by an X). **b** Nucleosome core particle (adapted from PDB:7U52)[44] with the 1-nt gap positions at SHL−5.5, SHL−4.5, SHL−3.5, SHL−2.5, SHL−1.5, SHL−4.5+2, SHL−4.5+3, and SHL−4.5+4 labeled. **c** Pol β nucleotide insertion rates ($k_{obs}$) for Gap-DNA and the eight Gap-NCPs obtained from the single-turnover kinetic analysis. The kinetic parameter $k_{obs}$ was not quantified (NQ) for Gap-NCP−1.5 due to minimal product formation (~10%). See Supplementary Fig. 1b for associated data. **d** Amplitudes of Pol β product formation for Gap-DNA and the eight Gap-NCPs obtained from the single-turnover kinetic analysis. See Supplementary Fig. 1b for associated data. **e** Apparent binding affinities ($K_{d, app}$) of Pol β for ND-NCP and the eight Gap-NCPs obtained from the EMSA analysis. See Supplementary Fig. 2 for associated data. All data points in this figure represent the mean ± standard error of the mean from three independent replicate experiments. All source data in this figure are provided as a Source data file.

## Pol β catalyzes position-dependent nucleotide insertion in the nucleosome

To determine the efficiency of Pol β during gap-filling DNA synthesis in the nucleosome, we performed single-turnover pre-steady state enzyme kinetics (STK) for Pol β and the eight Gap-NCPs (Fig. 1c, d, Supplementary Table 1, and Supplementary Fig. 1b). These experiments enabled us to monitor the ability of Pol β (in vast excess of the Gap-NCP substrate) to catalyze a single nucleotide insertion into each Gap-NCP (i.e., conversion of substrate to product) as a function of time. The resulting data was fit to an exponential to determine the kinetic parameter $k_{obs}$, which represents the Pol β nucleotide insertion rate[45]. In addition, we also determined the amplitude of product formation from these experiments, which represents the maximum product formed during a single Pol β catalytic cycle (i.e., single-turnover). As a control, we initially performed the STK analysis with a non-nucleosomal 147 bp Widom 601 DNA containing a 1-nt gap (non-nucleosomal Gap-DNA) and determined a $k_{obs}$ of $0.30 \pm 0.02\,s^{-1}$ (Fig. 1c, Supplementary Table 1, and Supplementary Fig. 1b). Notably, this value is similar to the $k_{obs}$ of $0.12\,s^{-1}$ previously reported for Pol β using a shorter 31 bp Gap-DNA substrate[46], but slightly slower than previously reported maximum rate constants for Pol β nucleotide insertion ($k_{pol}$)[47–49]. To determine how the translational position of the 1-nt gap affects Pol β nucleotide insertion, we performed STK analysis on Gap-NCP−5.5, Gap-NCP−4.5, Gap-NCP−3.5, Gap-NCP−2.5, and Gap-NCP−1.5. When the 1-nt gap was positioned proximal to the nucleosome entry/exit site at SHL−5.5, the Pol β nucleotide insertion rate was $0.12 \pm 2.7 \times 10^{-3}\,s^{-1}$, which is a modest 2.5x decrease from non-nucleosomal Gap-DNA (Fig. 1c, Supplementary Table 1, and Supplementary Fig. 1b). As the translational position of the 1-nt gap moved closer to the nucleosome dyad, we observed a stepwise decrease in Pol β activity resulting in nucleotide insertion rates of $0.01 \pm 9.1 \times 10^{-4}\,s^{-1}$ for Gap-NCP−4.5, $4.6 \times 10^{-3} \pm 1.6 \times 10^{-4}\,s^{-1}$ for Gap-NCP−3.5, and $1.8 \times 10^{-3} \pm 1.4 \times 10^{-4}\,s^{-1}$ for Gap-NCP−2.5, which represents a 33x, 65x, and 167x reduction compared to non-nucleosomal Gap-DNA, respectively (Fig. 1c, Supplementary Table 1, and Supplementary Fig. 1b). Notably, we observed minimal Pol β product formation (>10%) during the kinetic time course for Gap-NCP−1.5, indicating Pol β is inefficient at performing nucleotide insertion near the nucleosome dyad (Fig. 1c, d, Supplementary Table 1, and Supplementary Fig. 1b).

To determine whether the rotational orientation of the 1-nt gap effects Pol β nucleotide insertion, we performed additional STK analysis with Gap-NCP−4.5+2, Gap-NCP−4.5+3, and Gap-NCP−4.5+4 (Fig. 1c, d, Supplementary Table 1, and Supplementary Fig. 1b). The STK analysis revealed similar nucleotide insertion rates across the occluded rotational orientations with a $k_{obs}$ of $4.7 \times 10^{-4} \pm 2.0 \times 10^{-4}$ for Gap-NCP−4.5+2, $3.9 \times 10^{-4} \pm 1.9 \times 10^{-4}\,s^{-1}$ for Gap-NCP−4.5+3, and $4.9 \times 10^{-4} \pm 5.1 \times 10^{-5}\,s^{-1}$ for Gap-NCP−4.5+4. This represents a 638x, 770x, and 612x reduction compared to non-nucleosomal Gap-DNA and a 21x, 26x, and 20x reduction compared to the solvent-exposed Gap-NCP−4.5, respectively. While the rate of nucleotide insertion was similar between the three occluded rotational positions, we observed differences in the final amplitude of product formation for Gap-NCP−4.5+2 ($60 \pm 4\%$), Gap-NCP−4.5+3 ($26 \pm 2\%$), and Gap-NCP−4.5+4 ($31 \pm 3\%$) compared to the solvent-exposed Gap-NCP−4.5 ($89 \pm 1\%$) at the same translational position (Fig. 1d and Supplementary Table 1). The differences in the amplitude of product formation indicate a decrease in the amount of productive Pol β:Gap-NCP nucleotide insertion complex as the 1-nt gap becomes further occluded by the histone octamer. Together, the STK analysis indicates Pol β can perform robust nucleotide insertion in the nucleosome, though the nucleotide insertion rate and the amount of enzymatically productive Pol β:Gap-NCP complex varies depending on the translational position of the 1-nt gap relative to the nucleosome dyad and the rotational orientation of the 1-nt gap relative to the histone octamer (i.e., gap accessibility).

To assess the ability of Pol β to engage nucleosomal 1-nt gaps and determine if 1-nt gap recognition is position-dependent, we determined the apparent binding affinities ($K_{d,\,app}$) for a nucleosome without DNA damage (ND-NCP) and the eight Gap-NCPs using electrophoretic mobility shift assays (Fig. 1e, Supplementary Table 1, Supplementary Fig. 2). The $K_{d,\,app}$ of Pol β for the ND-NCP was $217 \pm 14\,nM$, indicating Pol β can bind the nucleosome even in the absence of a 1-nt gap. The $K_{d,\,app}$ of Pol β for each of the eight individual Gap-NCPs ranged from $51 \pm 3\,nM$ to $143 \pm 9\,nM$, which are a subtle ~4.3x to 1.5x increases in affinity compared to the ND-NCP. This indicates that Pol β engages the nucleosome with nanomolar affinity and has a modest specificity for nucleosomal 1-nt gaps at different translational positions and rotational orientations in the nucleosome. Though we did observe small differences in Pol β apparent binding affinity for the 1-nt gaps at each translational and rotational position in the nucleosome, these differences are orders of magnitude smaller than the reduction in $k_{obs}$ (compare Fig. 1c, e and Supplementary Table 1). This indicates the differences in Pol β nucleotide insertion rate observed at the varying translational positions and rotational orientations in the nucleosome are not the result of large differences in substrate binding affinity.

## Structural basis of 1-nt gap recognition in the nucleosome by Pol β

To understand the structural mechanism used by Pol β to recognize 1-nt gaps in the nucleosome, we generated a pre-catalytic Pol β-Gap-NCP−4.5 complex that was stabilized via glutaraldehyde cross-linking (see "Methods") and subjected the complex to single particle analysis (Supplementary Fig. 3). The cryo-EM dataset resulted in a Gap-NCP−4.5 structure without Pol β bound and a Pol β-Gap-NCP−4.5 structure where Pol β is in a pre-catalytic conformation poised for incoming dNTP binding and gap-filling DNA synthesis. The Gap-NCP−4.5 and Pol β-Gap-NCP−4.5 structures were resolved to a global resolution of 3.1 Å and 3.3 Å, respectively (Supplementary Figs. 4, 5). Importantly, the local resolution of Pol β in the Pol β-Gap-NCP−4.5 structure was 4−6 Å (Supplementary Fig. 5), which was sufficient to dock a previously determined high-resolution crystal structure of Pol β (PDB: 3ISB)[50].

In the Pol β-Gap-NCP−4.5 structure, Pol β engages the nucleosomal DNA with a ~10 bp footprint that spans from SHL−4 to SHL−5 and buries ~1033 Å² of nucleosome surface area (Fig. 2a, b). Notably, Pol β only interacts with the nucleosomal DNA and does not make direct contacts with the core histone octamer. The Pol β-nucleosomal DNA binding footprint is mediated by an extensive network of non-specific interactions between the lyase domain, polymerase domain, and the backbone phosphates of the nucleosomal DNA (Fig. 2c, d). The interaction between Pol β and the nucleosomal DNA is centered on the 1-nt gap, where the enzyme is engaged with the terminal 5′-phosphate (dC-119) and the primer terminal 3′-OH (dT-117). The 5′-phosphate is stabilized by several residues in the lyase domain (K35, Y39, K68, and K72) that play key roles in catalyzing the Pol β 5′-dRP lyase activity (Fig. 2e)[21,22,47,51]. In addition to the direct contacts with the 5′-phosphate, H34 of the lyase domain stacks onto the terminal base pair on the 5′-phosphate side of the 1-nt gap, allowing the enzyme to open and separate the 1-nt gap[28,52]. This opening of the 1-nt gap facilitates the positioning of the primer terminal 3′-OH into the polymerase domain in close proximity to the catalytic triad (D190, D192, and D256) of the polymerase active site (Fig. 2f). While the local resolution of the polymerase domain precludes exact positioning of active site side chains (Supplementary Fig. 5), the proximity of the primer terminus to the catalytic triad strongly suggest Pol β is poised for incoming nucleotide binding and subsequent nucleotide insertion. Consistent with this conclusion, the overall conformation of Pol β and the nucleosomal DNA is virtually identical to that previously

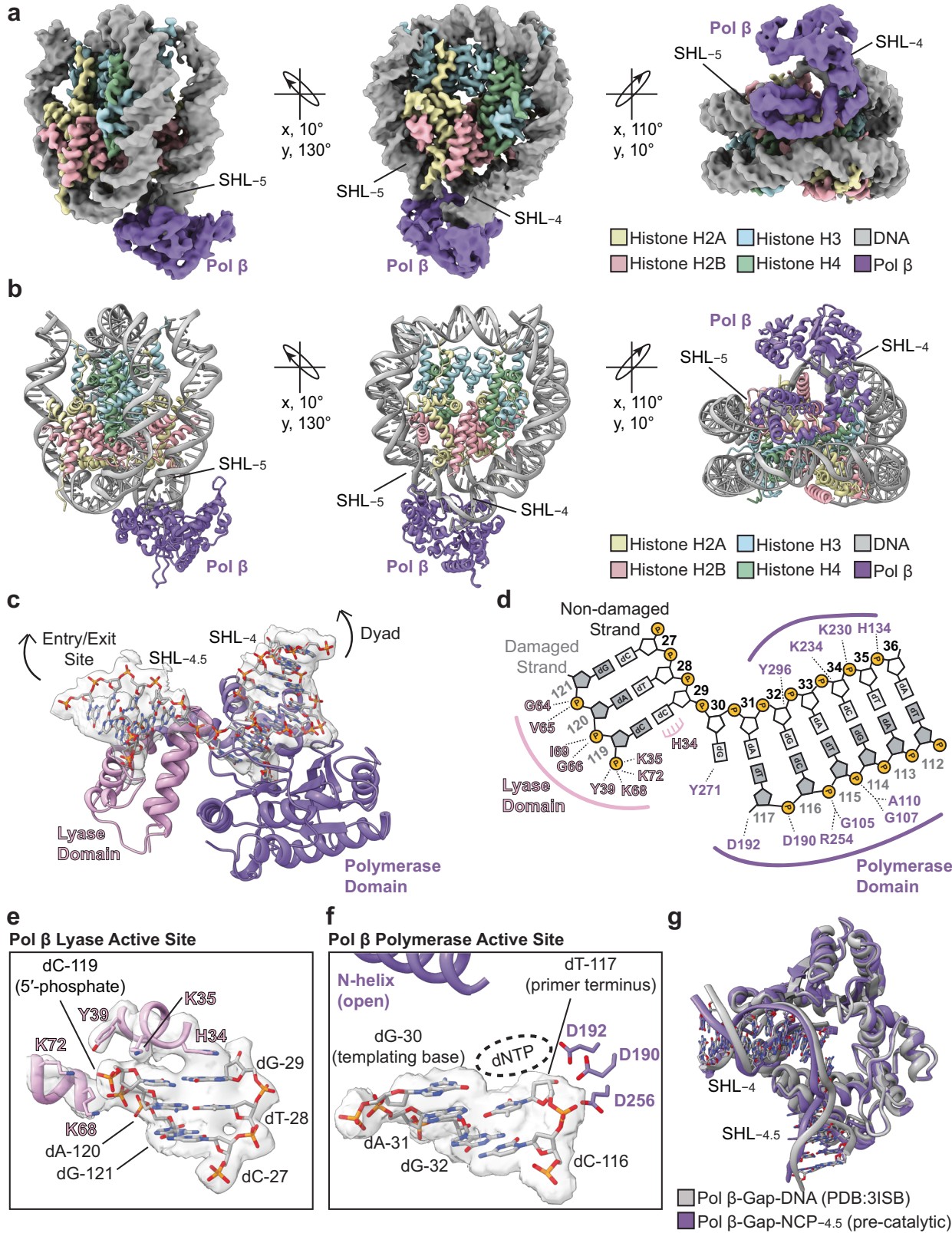

**e** **Pol β Lyase Active Site**

**f** **Pol β Polymerase Active Site**

observed in the high-resolution structure of the pre-catalytic Pol β-Gap-DNA complex (PDB: 3ISB)[50] (Fig. 2g). This structural comparison also strongly suggests that Pol β uses the same general mechanism for 1-nt gap recognition and gap-filling DNA synthesis in nucleosomal and non-nucleosomal DNA[28,50].

During 1-nt gap recognition, Pol β induces large structural distortions in the nucleosomal DNA that enables the enzyme to engage

the 1-nt gap (Fig. 3a and Supplementary Fig. 6a). The 5′-phosphate and primer terminal 3′-OH undergo substantial ~17 Å and ~27 Å movements compared to their position in the Gap-NCP−4.5 structure without Pol β bound (Fig. 3a). This movement of the DNA ends uncouples the 5′-phosphate and primer terminal 3′-OH to reposition them into the lyase and polymerase active sites, respectively (Fig. 2e, f). The repositioning of the 5′-phosphate is accomplished by pulling and rotating the

**Fig. 2 | Structural basis of 1-nt gap recognition in the nucleosome by Pol β.**
**a** Composite cryo-EM map of the pre-catalytic Pol β-Gap-NCP−4.5 complex in three different orientations with the cryo-EM model shown in (**b**) in the same three orientations. **c** Focused view of the Pol β-nucleosomal DNA binding interface at SHL −4.5. The segmented cryo-EM density for the nucleosomal DNA is shown as a transparent gray surface. **d** Contact map schematic showing the interactions between Pol β and the nucleosomal DNA identified using PLIP[96]. Pol β lyase domain residues and polymerase domain residues that interact with the nucleosomal DNA are labeled in pink and purple, respectively. DNA nucleobases are labeled and abbreviated dA (deoxyribose adenine), dG (deoxyribose guanine), dC (deoxyribose cytosine), or dT (deoxyribose thymine). **e** Focused view of the Pol β lyase domain

active site with active site residues shown as pink sticks. DNA nucleobases are labeled and abbreviated dA (deoxyribose adenine), dG (deoxyribose guanine), dC (deoxyribose cytosine), or dT (deoxyribose thymine). The segmented cryo-EM density for the nucleosomal DNA and Pol β lyase domain active site residues is shown as a transparent gray surface. **f** Focused view of the Pol β polymerase domain active site. Key polymerase domain active site residues are shown as purple sticks. DNA nucleobases are labeled and abbreviated dA (deoxyribose adenine), dG (deoxyribose guanine), dC (deoxyribose cytosine), or dT (deoxyribose thymine). The segmented cryo-EM density for the nucleosomal DNA is shown as a transparent gray surface. **g** Structural comparison of the pre-catalytic Pol β-Gap-NCP−4.5 complex (purple) and the pre-catalytic Pol β-Gap-DNA complex (gray, PDB:3ISB)[50].

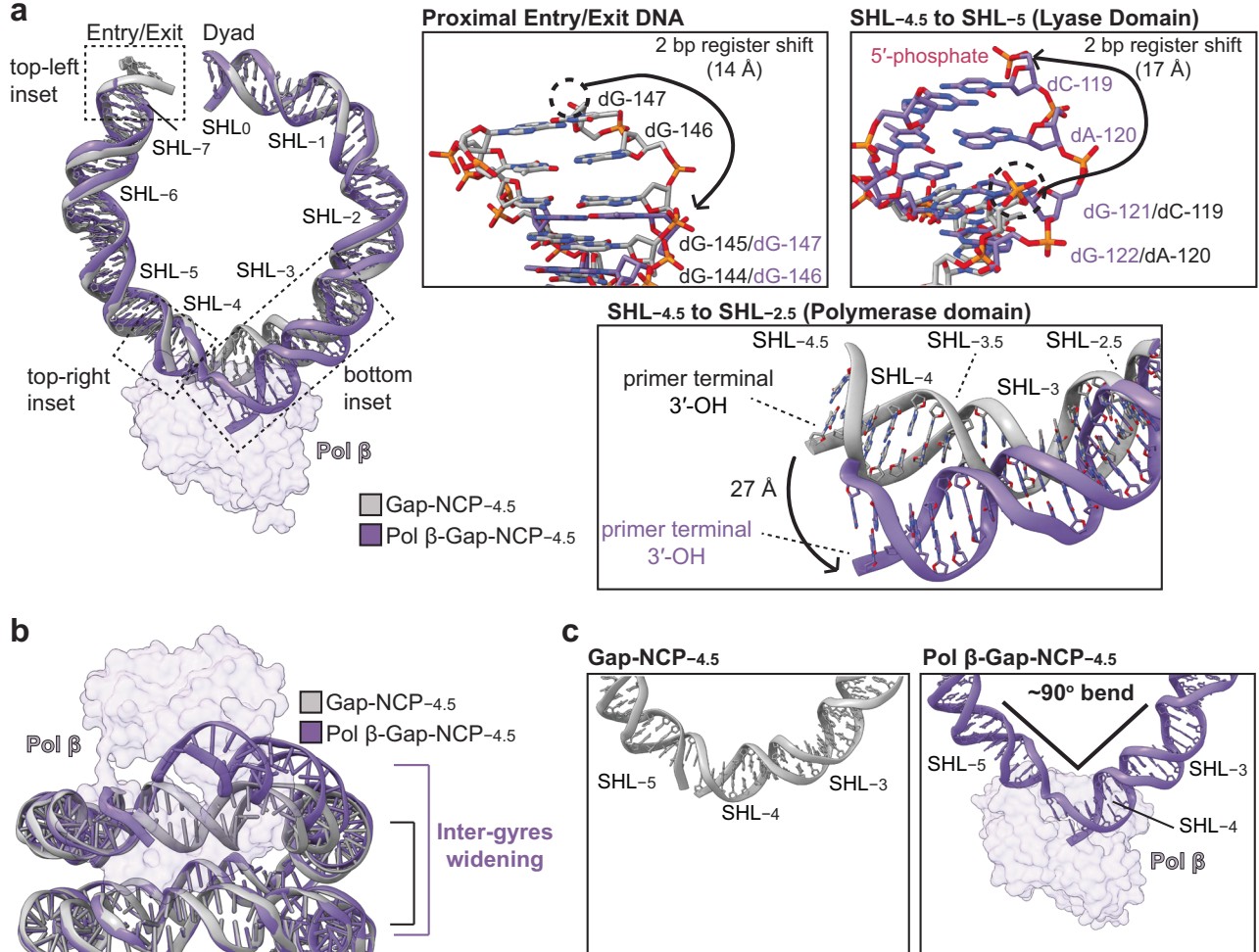

**Fig. 3 | Pol β induces structural distortions in the nucleosomal DNA during 1-nt gap recognition. a** Structural comparison of the nucleosomal DNA in the Gap-NCP −4.5 (gray) and the pre-catalytic Pol β-Gap-NCP−4.5 complex (purple). Pol β is shown as a transparent purple surface. DNA nucleobases are labeled and abbreviated dA (deoxyribose adenine), dG (deoxyribose guanine), dC (deoxyribose cytosine), or dT (deoxyribose thymine). Focused views of regions in the nucleosomal DNA that undergo structural changes upon 1-nt gap recognition by Pol β are shown as insets. These structural changes include the 2 bp register shift in the

nucleosomal DNA from the proximal entry/exit site (top-left inset) that repositions the 5′-phosphate into the lyase active site (top-right inset), and the lifting of the nucleosomal DNA directly off the histone octamer into the polymerase active site (bottom inset). **b** Structural comparison of the nucleosomal DNA in the Gap-NCP −4.5 (gray) and the pre-catalytic Pol β-Gap-NCP−4.5 complex (purple) from a side (inter-gyres) view. Pol β is shown as a transparent purple surface. **c** Focused views of the nucleosomal DNA from SHL−2.5 to SHL−5.5 in the Gap-NCP−4.5 (left, gray) and Pol β-Gap-NCP−4.5 complex (right, purple).

nucleosomal DNA from the proximal entry/exit site, which alters the rotational register of the nucleosomal DNA by 2 bp (Fig. 3a, top insets). In contrast, the movement of the primer terminal 3′-OH is accomplished by pulling the nucleosomal DNA directly off the histone octamer into the polymerase active site (Fig. 3a, bottom inset), which

disrupts the interactions between the H2A/H2B dimer and the nucleosomal DNA from SHL−3 to SHL−4 (Supplementary Fig. 6b). Additionally, we observed significant inter-gyres widening of the nucleosomal DNA at the Pol β binding site, termed nucleosome gaping[53], which is mediated by wedging of the Pol β lyase domain

between the two gyres of nucleosomal DNA (Fig. 3b). These combined structural changes in the nucleosomal DNA upon Pol β binding ultimately generate a ~90° bend in the nucleosomal DNA during 1-nt gap recognition that displaces ~35 bp of nucleosomal DNA in total (Fig. 3c and Supplementary Fig. 6a), a remarkable feat given the relatively small ~10 bp DNA binding footprint of Pol β. The substantial displacement of the nucleosomal DNA away from the histone octamer upon 1-nt gap recognition by Pol β likely explains the modest reduction in nucleotide insertion rate for solvent-exposed 1-nt gaps in nucleosomal DNA (near the entry/exit site) compared to non-nucleosomal DNA (Fig. 1c, Supplementary Table 1, and Supplementary Fig. 1b)[32–41].

## Sequential binding of the lyase and polymerase domains during 1-nt gap recognition in the nucleosome

During initial classification of the Pol β-Gap-NCP−4.5 cryo-EM dataset, we observed significant heterogeneity in Pol β and the surrounding nucleosomal DNA. Further 3D classification of the dataset resulted in two additional Pol β-Gap-NCP−4.5 structures that primarily differ in which Pol β domains are stabilized on the nucleosomal DNA and the extent of nucleosomal DNA distortion by Pol β (Fig. 4 and Supplementary Figs. 3, 7, 8). These structures represent intermediate snapshots of Pol β engaged with the nucleosomal DNA prior to the pre-catalytic state (state 3, Fig. 2), and we refer to these structures as the 5′-phosphate capture state (state 1) and the primer terminus capture state (state 2). The 5′-phosphate capture state (state 1) and the primer terminus capture state (state 2) were resolved to a global resolution of 3.3 Å and 3.3 Å, respectively (Supplementary Figs. 7 and 8). We envision these structures represent sequential snapshots of Pol β during 1-nt gap recognition in the nucleosome, which is based on the structural and biochemical analysis outlined below and prior studies that strongly suggest Pol β recognizes 1-nt gaps with the lyase domain prior to the polymerase domain[34,54–56].

In the 5′-phosphate capture state (state 1), the Pol β lyase domain is engaged with the 5′-phosphate and interacts with the nucleosomal DNA on the proximal entry/exit side of the 1-nt gap (Fig. 4a, b). Interestingly, the Pol β polymerase domain was poorly resolved in the 5′-phosphate capture state, with the exception of residues 88–146, and the remainder of the polymerase domain was only visible at significantly lower threshold (Fig. 4a and Supplementary Fig. 9a). The poorly resolved polymerase domain is likely the result of significant conformational heterogeneity and suggests the polymerase domain is not stably engaged with the primer terminus of the nucleosomal DNA. In the primer terminus capture state (state 2), the lyase domain remains engaged with the 5′-phosphate and the nucleosomal DNA on the proximal entry/exit side of the 1-nt gap (Fig. 4a, c). However, in contrast to the 5′-phosphate capture state, the polymerase domain was well resolved and is engaged with the primer terminus of the nucleosomal DNA (Fig. 4a–c).

Comparison of the Gap-NCP−4.5 structure and the three Pol β-Gap-NCP−4.5 structural states reveal stepwise distortions in the nucleosomal DNA (Fig. 4d), which are consistent with the sequential engagement of the lyase and polymerase domains. The initial capture of the 5′-phosphate separates the 5′-phosphate and primer terminal 3′-OH, which undergo ~17 Å and ~20 Å movements away from the histone octamer, compared to their initial position in the Gap-NCP−4.5 structure without Pol β bound (Fig. 4d, left). These movement are mediated by wedging of the lyase domain between the two ends of the 1-nt gap, which repositions the 5′-phosphate into the lyase active site and exposes the primer terminal 3′-OH (Fig. 4b). The repositioning of the 5′-phosphate into the lyase active site is accomplished by pulling and rotating the nucleosomal DNA from the proximal entry/exit site, altering the rotational register of the nucleosomal DNA by 2 bp. Importantly, the repositioning of the 5′-phosphate is strikingly similar to that observed in the pre-catalytic state (Fig. 3a, top insets), and suggests that the lyase domain is fully engaged with the 5′-phosphate.

The subsequent transitions from the 5′-phosphate capture state (state 1) to the primer terminus capture state (state 2), and from the primer terminus capture state (state 2) to the pre-catalytic state (state 3) are mediated by additional structural changes in Pol β and the nucleosomal DNA. This primarily includes movements in the primer terminal DNA and primer terminal 3′-OH away from the histone octamer (Fig. 4d, middle and right). Together, the sequential structural distortions in the nucleosomal DNA from the 5′-phosphate capture state (state 1) to the pre-catalytic state (state 3) ultimately position the primer terminal 3′-OH into the polymerase active site, which poises Pol β for nucleotide binding and subsequent catalysis (Fig. 2e, f).

To further test the sequential mechanism for 1-nt gap recognition in the nucleosome, we determined the apparent binding affinity of the Pol β lyase domain (8 kDa domain, residues 1–87)[57] for a ND-NCP and Gap-NCP−4.5 using EMSAs (Fig. 4e and Supplementary Fig. 10). We reasoned that if capture of the 5′-phosphate is the initial step during 1-nt gap recognition in the nucleosome, the Pol β lyase domain (in isolation) should be sufficient for high affinity nucleosome binding and for 1-nt gap specificity. The EMSA analysis revealed the Pol β lyase domain (residues 1–87) binds the ND-NCP and Gap-NCP−4.5 with a $K_{d, app}$ of $451 \pm 64$ nM and $K_{d, app} = 125 \pm 9$ nM, respectively. The apparent binding affinity of the Pol β lyase domain for the ND-NCP ($K_{d, app} = 451 \pm 64$ nM) represents a subtle ~2x decrease in affinity for the ND-NCP compared to full-length Pol β ($K_{d, app} = 217 \pm 14$ nM). However, the apparent binding affinity of the Pol β lyase domain for Gap-NCP−4.5 ($K_{d, app} = 125 \pm 9$ nM) was strikingly similar to that determined for full-length Pol β and Gap-NCP−4.5 ($K_{d, app} = 98 \pm 4$), indicating that the Pol β lyase domain has similar 1-nt gap specificity in the nucleosome as full-length Pol β. Importantly, the ability of the Pol β lyase domain to bind the nucleosome with high affinity and with 1-nt gap specificity is consistent with our cryo-EM snapshots showing the lyase domain mediates initial nucleosome binding through capture of the 5′-phosphate (Fig. 4a–d).

## Pol β uses a conserved mechanism for 1-nt gap recognition at different positions in the nucleosome

The STK analysis indicates that damage accessibility in the nucleosome strongly dictates Pol β nucleotide insertion efficiency (Fig. 1c, Supplementary Table 1, and Supplementary Fig. 1b), suggesting that Pol β may use an alternative mechanism for engaging and/or catalyzing gap-filling DNA synthesis at different 1-nt gap positions in the nucleosome. To determine how Pol β engages 1-nt gaps at additional positions in the nucleosome, we generated pre-catalytic Pol β-Gap-NCP−5.5 and Pol β-Gap-NCP−3.5 complexes that were stabilized via glutaraldehyde cross-linking (see "Methods") and subjected these complexes to single particle analysis (Supplementary Figs. 11, 14). Importantly, Pol β has a higher and lower nucleotide insertion rate at SHL−5.5 and SHL−3.5 than that observed at SHL−4.5 (Fig. 1c and Supplementary Table 1), respectively, allowing us to directly probe how damage accessibility impacts 1-nt gap recognition. The cryo-EM datasets resulted in structures of the Gap-NCP−5.5, Pol β-Gap-NCP−5.5 complex, Gap-NCP−3.5, and Pol β-Gap-NCP−3.5 complex, which were resolved to global resolutions of 3.9 Å, 4.2 Å, 3.4 Å, and 4.3 Å, respectively (Fig. 5a, b, Supplementary Figs. 11–16, and Supplementary Table 3). Though the resolution of the maps for the Pol β-Gap-NCP−5.5 complex and Pol β-Gap-NCP−3.5 complex precludes accurate modeling of side chains, both maps were of sufficient quality to unambiguously dock Cα models of the nucleosome and Pol β (Supplementary Figs. 13, 16).

The general mechanism of nucleosome binding and 1-nt gap recognition by Pol β at SHL−5.5 and SHL−3.5 are similar to that observed at SHL−4.5 (Figs. 5c–e and 2c, d), suggesting these structures represent Pol β poised for incoming nucleotide binding and catalysis. Despite the similar mode of nucleosome binding and 1-nt gap recognition, we observed stark differences in the displacement of the nucleosomal DNA by Pol β when the 1-nt gap was moved away from the

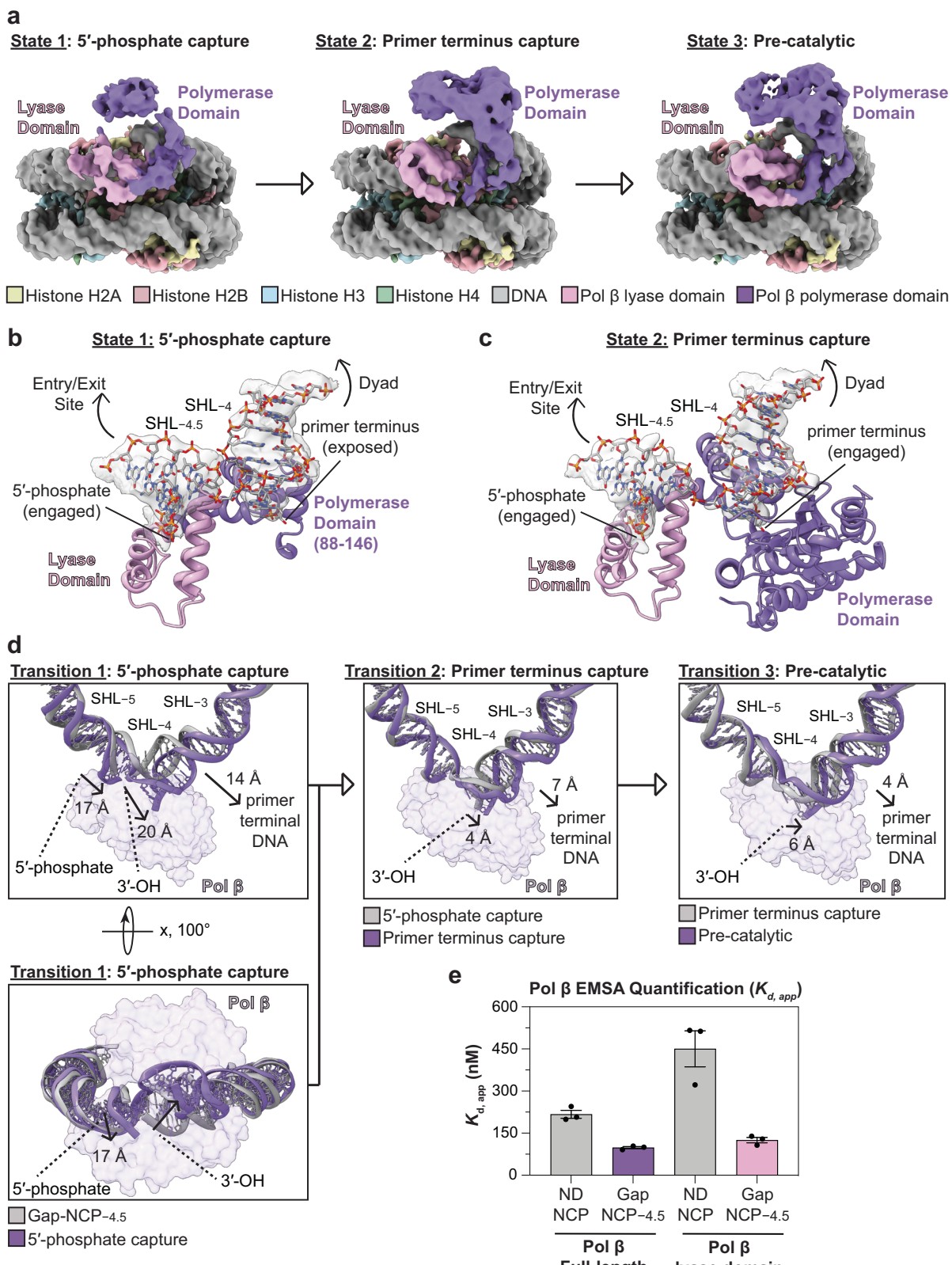

**a**
State 1: 5′-phosphate capture | State 2: Primer terminus capture | State 3: Pre-catalytic

Polymerase Domain — Lyase Domain

☐ Histone H2A  ☐ Histone H2B  ☐ Histone H3  ☐ Histone H4  ☐ DNA  ☐ Pol β lyase domain  ☐ Pol β polymerase domain

**b** State 1: 5′-phosphate capture

Entry/Exit Site — Dyad — SHL−4 — SHL−4.5 — primer terminus (exposed) — 5′-phosphate (engaged) — Polymerase Domain (88-146) — Lyase Domain

**c** State 2: Primer terminus capture

Entry/Exit Site — Dyad — SHL−4 — SHL−4.5 — primer terminus (engaged) — 5′-phosphate (engaged) — Lyase Domain — Polymerase Domain

**d**
Transition 1: 5′-phosphate capture
SHL−5 — SHL−3 — SHL−4 — 14 Å — primer terminal DNA — 17 Å — 20 Å — 5′-phosphate — 3′-OH — Pol β

Transition 2: Primer terminus capture
SHL−5 — SHL−3 — SHL−4 — 7 Å — primer terminal DNA — 4 Å — 3′-OH — Pol β
☐ 5′-phosphate capture
☐ Primer terminus capture

Transition 3: Pre-catalytic
SHL−5 — SHL−3 — SHL−4 — 4 Å — primer terminal DNA — 6 Å — 3′-OH — Pol β
☐ Primer terminus capture
☐ Pre-catalytic

x, 100°

Transition 1: 5′-phosphate capture
Pol β — 17 Å — 5′-phosphate — 3′-OH
☐ Gap-NCP−4.5
☐ 5′-phosphate capture

**e** Pol β EMSA Quantification ($K_{d, app}$)

$K_{d, app}$ (nM): 600, 450, 300, 150, 0

ND NCP | Gap NCP−4.5 | ND NCP | Gap NCP−4.5
Pol β Full-length | Pol β lyase domain

nucleosome entry/exit site and towards the nucleosome dyad (i.e., from SHL−5.5/SHL−4.5 to SHL−3.5). At SHL−5.5, Pol β binding generated a substantial 90° bend in the nucleosomal DNA during 1-nt gap recognition (Fig. 5f), which is almost identical to that observed in the Pol β-Gap-NCP−4.5 structure (Fig. 3c). Pol β also generated a -90° bend in the nucleosomal DNA during 1-nt gap recognition at SHL−3.5 (Fig. 5g). However, we did not observe the terminal 21 bp of

nucleosomal DNA at the proximal entry/exit site, indicating the Pol β-induced 90° bend in the nucleosomal DNA at SHL−3.5 is accommodated through unwrapping of the nucleosomal DNA at the proximal entry/exit site (Fig. 5h). Importantly, this unwrapping of the proximal entry/exit site nucleosomal DNA was not observed in the structures of Pol β bound to a 1-nt gap at SHL−5.5 or SHL−4.5. Together, these structures indicate that Pol β uses the same general

**Fig. 4 | Sequential binding of the lyase and polymerase domains during 1-nt gap recognition in the nucleosome. a** Composite cryo-EM maps of the three distinct Pol β-Gap-NCP−4.5 complexes including the 5′-phosphate capture state (state 1), primer terminus capture state (state 2), and pre-catalytic state (state 3). **b** Focused view of the Pol β-nucleosomal DNA binding interface at SHL−4.5 in the 5′-phosphate capture state (state 1). **c** Focused view of the Pol β-nucleosomal DNA binding interface at SHL−4.5 in the primer terminus capture state (state 2). **d** Structural comparison of the nucleosomal DNA from SHL−2.5 to SHL−5.5 during each individual transition from the apo Gap-NCP−4.5 to the pre-catalytic state (state 3). A

second view (bottom inset) of the transition between the Gap-NCP−4.5 structure and the 5′-phosphate capture structure (state 1) is shown to better highlight the initial movement of the 5′-phosphate. Pol β is shown as a transparent purple surface. **e** Quantification of the apparent binding affinities ($K_{d, app}$) of full-length Pol β and the Pol β lyase domain (residues 1–87) for ND-NCP and Gap-NCP−4.5 obtained from the EMSA analysis. The data points represent the mean ± standard error of the mean from three independent replicate experiments. The apparent binding affinities ($K_{d, app}$) of full-length Pol β are the same as in Fig. 1e. See Supplementary Figs. 2 and 10 for associated data. All source data in this figure are provided as a Source data file.

mechanism for 1-nt gap recognition and gap-filling DNA synthesis at different positions in the nucleosome, though additional structural changes in the nucleosomal DNA outside of the Pol β binding footprint are likely required as the 1-nt gap becomes less accessible (e.g., unwrapping of the proximal entry/exit DNA).

## Discussion

Single-strand breaks (SSBs) are one of the most prevalent forms of genomic DNA damage, which must be efficiently repaired in the context of chromatin to maintain genome stability. Despite the importance in understanding how SSBs are repaired in chromatin, mechanistic insight into their repair in the context of the nucleosome has been elusive. Our work has unraveled the molecular basis of gap-filling DNA synthesis in the nucleosome by Pol β, significantly enhancing our understanding of 1-nt gap processing in chromatin during SSBR and BER. Our initial kinetic analysis identified that Pol β has robust nucleotide insertion efficiency in the nucleosome, particularly when 1-nt gaps are solvent exposed near the nucleosome entry/exit site. Our cryo-EM structure of the pre-catalytic Pol β-Gap-NCP−4.5 complex provides a structural basis for this robust enzymatic activity. Pol β binds the nucleosomal 1-nt gap by generating multiple structural distortions in the nucleosome that displaces the nucleosomal DNA from the histone octamer (Figs. 2 and 3). This combination of Pol β-induced structural changes in the nucleosomal DNA, herein referred to as DNA sculpting, ultimately positions the primer terminal 3′-OH into the polymerase active site for catalysis. Interestingly, the DNA sculpting mechanism used by Pol β during 1-nt gap recognition in the nucleosome, which impacts ~35 bp of nucleosomal DNA, is more extensive than the localized DNA sculpting previously observed for the upstream BER/SSBR enzymes alkyladenine DNA glycosylase (AAG)[58], 8-oxoguanine DNA glycosylase-1 (OGG1)[3,59], and APE1[44] bound to their cognate DNA lesions in the nucleosome. Importantly, this may explain why the enzymatic activity of upstream BER enzymes (i.e., DNA glycosylases and APE1)[44,60–62] is generally less sensitive to nucleosome structure compared to the downstream BER enzymes (i.e., Pol β and DNA Ligase IIIa)[32–41,63].

Prior biochemical studies on non-nucleosomal DNA indicate that Pol β uses a stepwise mechanism to engage 1-nt gaps[34,54–56]. During this stepwise mechanism, Pol β initially engages the 1-nt gap by capturing the 5′-phosphate using the lyase domain. After capture of the 5′-phosphate, the polymerase domain then engages the primer terminus and positions the primer terminus into the polymerase active site. Consistent with this model, we determined two Pol β-Gap-NCP−4.5 structural intermediates that precede the pre-catalytic state, termed the 5′-phosphate capture state (state 1) and the primer terminus capture state (state 2). These structures indicate Pol β also likely uses a stepwise mechanism for 1-nt gap recognition in the nucleosome, where capture of the 5′-phosphate by the lyase domain precedes capture of the primer terminus by the polymerase domain during the transition to the pre-catalytic state. Though it is possible that these structural snapshots may represent artifacts arising from glutaraldehyde cross-linking of the Pol β-Gap-NCP−4.5 complex, our supporting biochemical assays indicate that the Pol β lyase domain mediates initial nucleosome binding and 1-nt gap recognition, which is consistent with the observations made in these structural snapshots.

The importance of the Pol β lyase domain during initial nucleosome binding and 1-nt gap recognition suggests the lyase domain could play an important role during Pol β search for 1-nt gaps in chromatin[55], which could partially explain the necessity of this small domain for efficient recruitment of Pol β to sites of DNA damage[56] and cellular viability in response to exogenous DNA damaging agents[64].

Our systematic kinetic analysis identified that Pol β nucleotide insertion efficiency is position-dependent in the nucleosome. This position-dependent activity is dictated by the translational position of the 1-nt gap (entry/exit site > dyad) and the rotational orientation of the 1-nt gap (solvent-exposed > histone-occluded). It's important to note that genomic sequences that are less stable in positioning the histone octamer compared to the Widom 601 DNA may impact the magnitude of this position-dependent enzymatic activity. In addition, we also point out that the Widom 601 sequence in our experiments was not altered to match the templating base and incoming dNTP at all eight Gap-NCP positions (Supplementary Table 1 and Supplementary Data 1), which could subtly impact our kinetic analysis[65,66]. Nevertheless, the trends in Pol β nucleotide insertion rate we observed at different translational positions and rotational orientations in the nucleosome are generally consistent with prior observations made for Pol β[32–41] and upstream BER/SSBR enzymes on defined nucleosome substrates[44,60–62].

Why is the enzymatic activity Pol β so sensitive to the translational position and rotational orientation of the 1-nt gap in the nucleosome? Our cryo-EM structures of Pol β bound to nucleosomal 1-nt gaps at SHL−5.5, SHL−4.5, and SHL−3.5 provide a plausible rationale for this position-dependent enzymatic activity. In these structures, Pol β uses the same general global DNA sculpting mechanism to engage the 1-nt gap and position the primer terminal 3′-OH into the polymerase active site. However, we observed accessibility-dependent differences in nucleosome distortion during 1-nt gap recognition in the nucleosome. At SHL−3.5 (i.e., the least accessible translational position), Pol β induces unwrapping of the proximal entry/exit nucleosomal DNA during 1-nt gap recognition at SHL−3.5, which was not observed during recognition of 1-nt gaps at SHL−5.5 and SHL−4.5 (i.e., more accessible translational positions). This observation at SHL−3.5 is likely due to the inability of the nucleosome to accommodate the localized 90° bend surrounding the Pol β binding site while simultaneously maintaining the canonical trajectory of the nucleosomal DNA around the histone octamer, which could be overcome through unwrapping of the nucleosomal DNA at the proximal entry/exit site. Though this is a single example of accessibility-dependent differences in nucleosome structure upon 1-nt gap recognition by Pol β, we envision that similar structural distortions in the nucleosomal DNA are likely required to accommodate the 90° bend induced by Pol β at other less accessible 1-nt gap translational positions in the nucleosome (e.g., from the dyad to SHL−3.5). Notably, this would readily explain with the stepwise decrease in Pol β nucleotide insertion rate as the position of the solvent-exposed 1-nt gaps get closer to the nucleosome dyad (Fig. 1, Supplementary Fig. 1, and Supplementary Table 1). Though our work does not provide direct structural insight into the reduction in Pol β nucleotide insertion rate and enzymatically productive Pol β:Gap-NCP complex at histone-occluded rotational orientations, we envision that Pol β is also unable to efficiently sculpt the nucleosomal DNA to

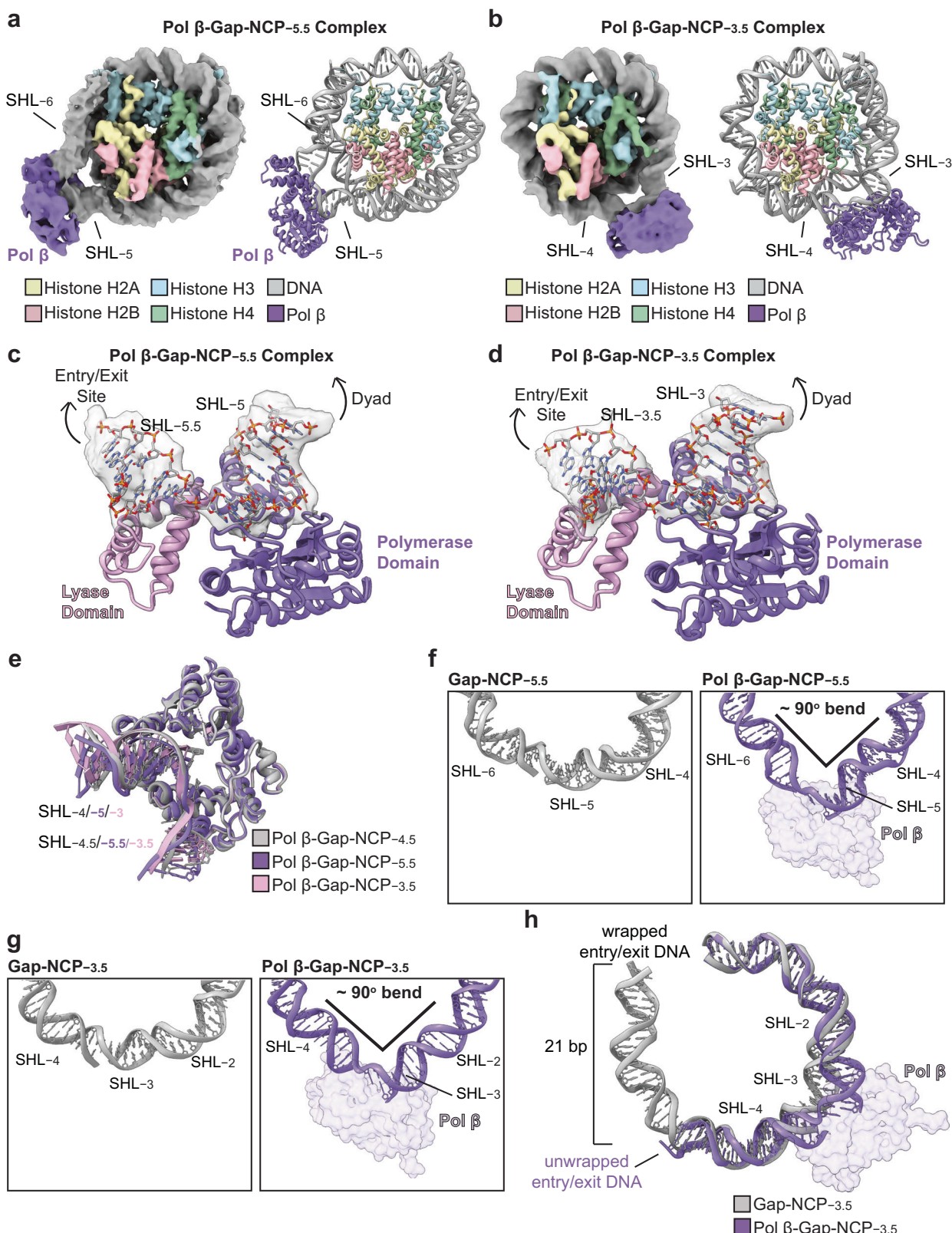

**a** Pol β-Gap-NCP_{-5.5} Complex

**b** Pol β-Gap-NCP_{-3.5} Complex

Histone H2A | Histone H3 | DNA
Histone H2B | Histone H4 | Pol β

**c** Pol β-Gap-NCP_{-5.5} Complex

**d** Pol β-Gap-NCP_{-3.5} Complex

**e**

Pol β-Gap-NCP_{-4.5}
Pol β-Gap-NCP_{-5.5}
Pol β-Gap-NCP_{-3.5}

**f** Gap-NCP_{-5.5} | Pol β-Gap-NCP_{-5.5}

~ 90° bend

**g** Gap-NCP_{-3.5} | Pol β-Gap-NCP_{-3.5}

~ 90° bend

**h**

wrapped entry/exit DNA
21 bp
unwrapped entry/exit DNA

Gap-NCP_{-3.5}
Pol β-Gap-NCP_{-3.5}

position the primer terminal 3′-OH into the polymerase active site for catalysis at these positions without substantially clashing with the core histone octamer.

The results of our kinetic analysis indicate that 1-nt gaps in proximity to the nucleosome dyad or 1-nt gaps in histone-occluded rotational orientations are unable to be repaired efficiently by Pol β alone (Fig. 1, Supplementary Fig. 1, and Supplementary Table 1). This suggests that additional factors may be needed to accelerate the ability of Pol β to process 1-nt gaps in chromatin, particularly at inaccessible locations in the nucleosome. Indeed, a variety of BER/SSBR cofactors and chromatin modifying enzymes have been implicated in regulating chromatin structure in response to DNA damage to facilitate BER and SSBR[17,38,67,68]. For example, the critical BER and SSBR scaffolding protein X-ray repair cross complementing 1 (XRCC1) was shown to

**Fig. 5 | Pol β uses a conserved mechanism for 1-nt gap recognition at different positions in the nucleosome. a** Composite cryo-EM map and model of the pre-catalytic Pol β-Gap-NCP−5.5 complex. **b** Cryo-EM map and model of the pre-catalytic Pol β-Gap-NCP−3.5 complex. **c** Focused view of the Pol β-nucleosomal DNA binding interface at SHL−5.5 in the Pol β-Gap-NCP−5.5 complex structure. The segmented cryo-EM density for the nucleosomal DNA is shown as a transparent gray surface. **d** Focused view of the Pol β-nucleosomal DNA binding interface at SHL −3.5 in the Pol β-Gap-NCP−3.5 complex structure. The segmented cryo-EM density for the nucleosomal DNA is shown as a transparent gray surface. **e** Structural comparison of Pol β and the nucleosomal DNA in the pre-catalytic Pol β-Gap-NCP −3.5 (pink), Pol β-Gap-NCP−4.5 (gray), and Pol β-Gap-NCP−5.5 (purple) complexes. **f** Focused views of the nucleosomal DNA from SHL−3.5 to SHL−6.5 in the Gap-NCP −5.5 (left) and Pol β-Gap-NCP−5.5 complex (right). **g** Focused views of the nucleosomal DNA from SHL−1.5 to SHL−4.5 in the Gap-NCP−3.5 (left) and Pol β-Gap-NCP −3.5 complex (right). **h** Structural comparison of the nucleosomal DNA in the Gap-NCP−3.5 and the pre-catalytic Pol β-Gap-NCP−3.5 complex. Pol β is shown as a transparent purple surface in Fig. 5f–h.

accelerate the ability of Pol β to process 1-nt gaps at inaccessible positions in the nucleosome in vitro[36,63]. In addition, the histone ADP-ribosylation signaling cascade involving Poly (ADP-Ribose) Polymerases (PARPs)1-3[10,69–77] and Amplified in Liver Cancer 1 (ALC1)[78–82] has been implicated in facilitating BER and/or SSBR in vitro and in vivo by directly altering chromatin structure[81,83–85]. These additional cellular factors may enhance the ability of Pol β to process 1-nt gaps at inaccessible locations in the nucleosome, though future work is needed to dissect the interplay between Pol β and additional BER/SSBR regulators during the processing of 1-nt gaps in chromatin.

## Methods

### Purification of Pol β

The codon-optimized gene for wild-type full-length (FL) human Pol β was cloned into an untagged pET28a bacterial expression vector using GenScript. The rat Pol β lyase domain (8 kDa domain, residues 1–87) in a pRSET bacterial expression vector was a generous gift from the laboratory of Dr. Samuel Wilson (NIEHS/NIH)[57]. Importantly, the rat protein has been used extensively to study Pol β lyase domain function and differs from the human Pol β lyase domain by a single amino acid residue (A20 in rat and T20 in human)[56,57,86].

The Pol β proteins were transformed and expressed in BL21-CodonPlus (DE3)-RP *Escherichia coli (E. coli)* competent cells (Agilent). The transformed cells were grown at 37 °C to an $OD_{600}$ of 0.6 and Pol β expression induced using 0.1 mM isopropyl-b-D-thiogalactopyranoside (IPTG) for 16–18 h at 18 °C. The purification of Pol β proteins (human FL and rat lyase domain) were carried out as previously described[87]. In brief, the cells were lysed via sonication in a buffer containing 50 mM HEPES (pH 7.5), 50 mM NaCl, 1 mM DTT, and a cocktail of protease inhibitor (AEBSF, leupeptin, benzamidine, pepstatin A). The lysate was clarified via centrifugation at 24,000 RCF and loaded onto a HiTrap Heparin HP column (Cytiva) equilibrated with 50 mM HEPES (pH 7.5), 50 mM NaCl, and 1 mM DTT, and eluted in a buffer 50 mM HEPES (pH 7.5), 1 M NaCl, and 1 mM DTT. The Pol β proteins were further purified using a GE Resource S (Cytiva) equilibrated with 50 mM HEPES (pH 7.5), 50 mM NaCl, and 1 mM DTT, and eluted in a buffer 50 mM HEPES (pH 7.5), 1 M NaCl, and 1 mM DTT. The resulting Pol β proteins were combined and purified via size exclusion chromatography using a HiPrep Sephacryl S-200 16/60 HR (Cytiva) in a buffer containing 50 mM HEPES (pH 7.5) and 50 mM NaCl. The purity of the Pol β proteins were confirmed via denaturing SDS PAGE and the purified Pol β proteins stored long term at −80 °C. The concentration of Pol β (human FL and rat lyase domain) were determined spectrophotometrically by measuring the 280 nm absorbance using a NanoDrop One UV–Vis Spectrophotometer (Thermo Scientific). All Pol β protein concentrations used throughout the methods are total Pol β concentration, not active Pol β concentration.

### Preparation of oligonucleotides

All oligonucleotides used in this study were synthesized by Integrated DNA Technologies (Coralville, IA). A complete list of oligonucleotides can be found in Supplementary Data 1. All oligonucleotides were resuspended in a buffer containing 10 mM Tris (pH-7.5) and 1 mM EDTA. Complementary oligonucleotides were annealed in a buffer containing 10 mM Tris (pH-7.5) and 1 mM EDTA by heating to 90 °C for 5 min and subsequently cooling to 4 °C at a rate of −1 °C/s. The annealed oligonucleotides were stored short-term at 4 °C or long-term at −20 °C.

### Purification of recombinant human histones

The genes encoding human histone H2A (UniProt identifier: P0C0S8), H2B (UniProt identifier: P62807), H3 C110A (UniProt identifier Q71DI3), and H4 (Uniprot identifier: P62805) were previously cloned into a pET3a expression vector. Histone H2A, H3, and H4 were individually transformed and expressed in T7 Express lysY/Iq competent *E. coli* cells (New England BioLabs). Histone H2B was transformed and expressed in BL21-CodonPlus(DE3)-RIPL *E. coli* cells (Agilent). Each histone was grown in M9 minimal media supplemented with a 1.0% vitamin solution at 37 °C to an $OD_{600}$ of 0.4, and expression induced with a 0.2 mM (histone H4) or 0.4 mM (histone H2A, H2B, and H3) IPTG for 3 h (histone H2A and H2B) or 4 h (Histone H3 and H4). The cells were harvested via centrifugation and cell pellets stored long term at −80 °C. The purification of individual histones was performed using previously established protocols[88,89]. In brief, the cell pellets were lysed via sonication in a buffer containing 50 mM Tris (pH 7.5), 100 mM NaCl, 1 mM benzamidine, 1 mM DTT, and 1 mM EDTA. The cell lysate was clarified via centrifugation and the resulting pellet washed three times with a buffer containing 50 mM Tris (pH 7.5), 100 mM NaCl, 1 mM benzamidine, 1 mM DTT, 1 mM EDTA, and 1% Triton X-100. The histones were extracted from inclusion bodies under denaturing condition (6 M Guanidinium-HCl) and purified using anion-exchange and cation-exchange chromatography under gravity flow. The purified histones were dialyzed five times against $H_2O$, lyophilized, and stored at −20 °C.

### Preparation of H2A/H2B Dimer and H3/H4 tetramer

Prior to nucleosome assembly, H2A/H2B dimers and H3/H4 tetramers were prepared. Each individual histone was resuspended into a buffer containing 20 mM Tris (pH 7.5), 6 M Guanidinium-HCl, and 10 mM DTT. For the H2A/H2B dimer, the resuspended histone H2A and H2B proteins were mixed in a 1:1 molar ratio and dialyzed three times against a buffer containing 20 mM Tris (pH 7.5), 2 M NaCl, and 1 mM EDTA at 4 °C. For the H3/H4 tetramer, the resuspended histone H3 and H4 proteins were mixed in a 1:1 molar ratio and dialyzed three times against a buffer containing 20 mM Tris (pH 7.5), 2 M NaCl, and 1 mM EDTA at 4 °C. The refolded H2A/H2B dimer and H3/H4 tetramer were subsequently purified using a HiPrep Sephacryl S-200 16/60 HR gel filtration column (Cytiva) in a buffer containing 20 mM Tris (pH 7.5), 2 M NaCl, and 1 mM EDTA. Fractions containing pure H2A/H2B dimers and H3/H4 tetramer were combined and stored long term at −20 °C in 50% glycerol.

### Nucleosome assembly and purification

All NCPs were prepared using a modified salt-dialysis method[88,89]. To generate the NCPs, H2A/H2B dimer, H3/H4 tetramer, and DNA were mixed at a ratio of 2.2:1:1, respectively. Nucleosome assembly was accomplished by sequentially decreasing the concentration of NaCl

from 2 M NaCl to 1.5 M NaCl, to 1.0 M NaCl, to 0.75 M NaCl, to 0.5 M NaCl, to 0.25 M NaCl, and to 0 M NaCl over 24 h. The assembled NCPs were heat shocked at 37 °C to obtain uniform DNA positioning and purified by sucrose gradient ultracentrifugation (10–40% gradient). Final nucleosome purity and homogeneity was assessed via native PAGE (5%, 59:1 acrylamide:bis-acrylamide ratio). The NCPs were stored in 10 mM Tris (pH-7.5) and 1 mM EDTA at 4 °C.

### Single-turnover pre-steady state kinetic analysis

Single turnover kinetic assays were carried out by incubating Pol β (1 μM) with the respective NCP substrate (100 nM). Reactions were performed at 37 °C in a buffer containing 50 mM Tris-HCl (pH 7.4), 100 mM KCl, 0.1 mg/ml bovine serum albumin, 1 mM DTT, and 10 v/v glycerol. Reactions were initiated through the addition of divalent metal (10 mM $MgCl_2$) and the correct incoming nucleotide (50 μM dNTP). The reactions were quenched at varying time points through the addition of a 1:1 v/v ratio of formamide loading dye (100 mM EDTA, 80% deionized formamide, 0.25 mg/ml bromophenol blue, 0.25 mg/ml xylene cyanol, and 8 M urea). The quenched reactions were incubated at 95 °C for 5 min, and resolved via denaturing polyacrylamide gel (8%, 29:1 acrylamide to bis-acrylamide, 8 M urea). The substrate and product bands were visualized using an Amersham Typhoon RGB imager through excitation of the 6-FAM label located on the 5′-end of the I-strand (see Supplementary Data 1). The amounts of substrate and product at each time point were quantified with ImageJ and fit to a single exponential equation in GraphPad Prism using equation 1:

$$P = A(1 - e^{-kt})$$

Where $A$ is the amplitude of the reaction. And $k$ corresponds to the rate of nucleotide insertion ($k_{obs}$) by Pol β. Each time point represents the average of at least three independent experiments, and the error bars represent the standard deviation of the three independent experiments. The $k_{obs}$ is reported as the mean of the three independent replicate experiments ± the standard error of the mean. Quantifications associated with the single-turnover kinetic analysis are available in the source data file.

### Electrophoretic mobility shift assays (EMSAs)

EMSAs were performed by mixing 20 nM of the ND-NCP or each respective Gap-NCP with increasing concentrations of the full-length Pol β (50–5000 nM) or Pol β lyase domain (50–5000 nM) in a buffer containing 20 mM HEPES (pH 7.5), 100 mM NaCl, 1 mM EDTA, 1 mM DTT and 0.1 mg/ml bovine serum albumin for 20 min at 4 °C. Samples were mixed with an equal volume of 10% sucrose loading dye and run using native polyacrylamide gel (5%, 59:1 acrylamide:bis-acrylamide) electrophoresis in a 0.2x TBE buffer for 45 min at 4 °C. The EMSAs were visualized via the 6-FAM label (located on the 5′-end of the I-strand) using an Amersham Typhoon RGB Imager. The disappearance of the free NCP band was quantified using ImageJ and fit in GraphPad Prism to a one site binding model accounting for ligand depletion using equation 2:

$$AB = \frac{\left(A_T + B_T + K_{D,app}\right) - \sqrt{\left(A_T + B_T + K_{D,app}\right)^2 - 4(A_T B_T)}}{2}$$

Where $A_T$ is the Pol β concentration, $B_T$ is the NCP concentration, and $AB$ is the concentration of the Pol β-NCP complex. Each experimental point represents the average of at least three independent experiments, and the error bars represent the standard deviation of the three independent experiments. The $K_{d,app}$ is reported as the mean of the three independent replicate experiments ± the standard error of the mean. Quantifications associated with the EMSA analysis are available in the source data file.

### Cryo-EM sample and grid preparation

Pol β-Gap-NCP complexes were generated by incubating Pol β (4.0–12.0 μM) with each respective Gap-NCP (2.0–6.0 μM) at a 2:1 molar ratio in a buffer containing 25 mM HEPES (pH 7.5), 50 mM NaCl, 1 mM TCEP, and 5 mM EDTA for 10 min on ice. Following incubation, the reactions were cross-linked with glutaraldehyde (0.1%) for 20 min on ice, and the samples immediately loaded onto a Superdex S200 Increase 10/300 GL (Cytiva) equilibrated in a buffer containing 25 mM HEPES (pH 7.5), 50 mM NaCl, 1 mM TCEP, and 5 mM EDTA. Fractions containing Pol β-Gap-NCP complexes were combined and concentrated to 1.25–1.50 μM. The final sample quality was assessed by native PAGE (5%, 59:1 acrylamide:bis-acrylamide ratio). Representative native PAGE gels of the cryo-EM samples can be found in Supplementary Figs. 3, 11, and 14.

Cryo-EM grids were generated by plunge-freezing into liquid ethane using an FEI Vitrobot Mark IV. Each sample (3 μL) was applied to quantifoil R2/2 300 mesh copper cryo-EM grids that were prepared via glow discharge. The grids were blotted for 1–3 s at 4 °C and 95% humidity before plunge-freezing in liquid ethane using an FEI Vitrobot Mark IV. The cryo-EM grids were clipped and stored in liquid nitrogen prior to screening and data collection.

### Cryo-EM data collection and processing

The Pol β-Gap-NCP−4.5 and Pol β-Gap-NCP−3.5 cryo-EM data collections were performed at the Pacific Northwest Center for Cryo-EM (PNCC) on a 300 kV TFS Krios cryo-TEM equipped with a Gatan K3 direct electron detector and BioContinuum energy filter using SerialEM. The Pol β-Gap-NCP−5.5 data collection was performed at the University of Colorado Boulder Krios Electron Microscopy facility (BioKEM) on a 300 kV TFS T Krios G3i cryo-TEM equipped with a Falcon 4 direct electron detector and Selectris energy filter using EPU. All data collection parameters can be found in Supplementary Table 2 and Supplementary Table 3.

All cryo-EM data was processed with cryoSPARC[90] using similar data processing workflows (complete data processing workflows can be found in Supplementary Figs. 3, 11, and 14). The micrographs were initially imported into cryoSPARC, a correction for beam-induced drift was performed using cryoSPARC patch motion correction, and contrast transfer function (CTF) fit using cryoSPARC patch CTF-estimation. Manual curation was then performed to exclude micrographs of poor quality. Initial particle picks were performed using cryoSPARC picking to generate templates for automated template picking. The particles were then subjected to multiple rounds of 2D classification (at least 2 rounds) to generate the final particle stacks. Ab initio models were then generated using the final particle stacks, and heterogeneous refinement(s) performed to separate Gap-NCP and Pol β-Gap-NCP complexes.

The Gap-NCP structures were further subjected to 3D classification using a mask for the entry/exit nucleosomal DNA (weakside DNA) to separate wrapped and partially unwrapped nucleosomes. After classification, the particle stacks for each NCP structure were re-extracted at full box size (600 pixel/Å), subjected to a local CTF refinement, and a final non-uniform refinement. The Gap-NCP maps were then subjected to B-factor sharpening using PHENIX autosharpen[91].

For the Pol β-Gap-NCP structures, further 3D classification was performed using a mask for Pol β and the surrounding nucleosomal DNA. After one or multiple rounds of 3D classification, the particle stacks for each Pol β-Gap-NCP structure were re-extracted at full box size (600 pixel/Å) and subjected to a local CTF refinement and a non-uniform refinement to generate consensus maps. To improve interpretability of the Pol β-Gap-NCP−4.5 maps and Pol β-Gap-NCP−5.5

maps, a local refinement (without particle subtraction) was performed using a focus mask encompassing Pol β and the surrounding nucleosomal DNA. Composite maps were then generated for each Pol β-Gap-NCP−4.5 and the Pol β-Gap-NCP−5.5 structure using PHENIX Combine focus maps. The global resolution for all reconstructions was determined using Fourier shell correlation (FSC) 0.143 cut off.

## Model building and refinement

The initial nucleosome model was generated from a previously determined structure of nucleosome containing an AP-site (PDB: 7U52)[44]. For the Pol β-Gap-NCP structures, an initial Pol β model was generated using the high-resolution X-ray crystal structure of Human Pol β (PDB: 3ISB)[50]. Initial models were rigid body docked into the Gap-NCP and Pol β-Gap-NCP maps using University of California San Francisco (UCSF) Chimera[92]. The models were then refined iteratively using PHENIX[91] and Coot[93] with protein and DNA secondary structures restraints. All models were validated using MolProbity[94] prior to deposition.

Model coordinates for Gap-NCP structures were deposited in the Protein Data Bank (PDB) under accession codes 9DWL for Gap-NCP −5.5, 9DWF for Gap-NCP−4.5, and 9DWJ for Gap-NCP−3.5. Consensus maps for the Gap-NCP structures were deposited in the Electron Microscopy Data Bank (EMDB) under accession codes EMD-47254 for Gap-NCP−5.5, EMD-47242 for Gap-NCP−4.5, and EMD-47252 for Gap-NCP−3.5.

Model coordinates for Pol β-Gap-NCP structures were deposited in the PDB under accession codes 9DWG for Pol β-Gap-NCP−4.5 (5′-phosphate capture, state 1), 9DWH for Pol β-Gap-NCP −4.5 (primer terminus capture, state 2), 9DWI for Pol β-Gap-NCP −4.5 (pre-catalytic, state 3), 9DWM for Pol β-Gap-NCP−5.5, and 9DWK for Pol β-Gap-NCP−3.5. Composite maps for the Pol β-Gap-NCP structures were deposited in the EMDB under accession codes EMD-47243 for Pol β-Gap-NCP−4.5 (5′-phosphate capture, state 1), EMD-47246 for Pol β-Gap-NCP−4.5 (primer terminus capture, state 2), EMD-47249 for Pol β-Gap-NCP−4.5 (pre-catalytic, state 3), and EMD-47255 for Pol β-Gap-NCP−5.5. Consensus maps for the Pol β-Gap-NCP structures were deposited in the EMDB under accession codes EMD-47244 for Pol β-Gap-NCP−4.5 (5′-phosphate capture, state 1), EMD-47247 for Pol β-Gap-NCP−4.5 (primer terminus capture, state 2), EMD-47250 for Pol β-Gap-NCP −4.5 (pre-catalytic, state 3), EMD-47256 for Pol β-Gap-NCP−5.5, and EMD-47253 for Pol β-Gap-NCP−3.5. Local refine maps for the Pol β-Gap-NCP structures were deposited in the EMDB under accession codes EMD-47245 for Pol β-Gap-NCP−4.5 (5′-phosphate capture, state 1), EMD-47248 for Pol β-Gap-NCP−4.5 (primer terminus capture, state 2), EMD-47251 for Pol β-Gap-NCP−4.5 (pre-catalytic, state 3), and EMD-47256 for Pol β-Gap-NCP−5.5. All map and model figures were generated using UCSF ChimeraX[95].

## Reporting summary

Further information on research design is available in the Nature Portfolio Reporting Summary linked to this article.

## Data availability

Atomic coordinates for the reported structures have been deposited with the Protein Data Bank under accession numbers 9DWG, 9DWF, 9DWH, 9DWI, 9DWK, 9DWL, 9DWM. All cryo-EM maps are available from the Electron Microscopy Data Bank under accession numbers EMD-47242, EMD-47243, EMD-47244, EMD-47245, EMD-47246, EMD-47247, EMD-47248, EMD-47249, EMD-47250, EMD-47251, EMD-47252, EMD-47253, EMD-47254, EMD-47255 and EMD-47256. Atomic coordinates for the initial nucleosome model and the Pol β-Gap-DNA model were obtained from the Protein Data bank under accession numbers 7U52 and 3ISB, respectively. The data from the single-turnover enzyme kinetics and EMSA experiments generated in this study are available in the Supplementary Information file and the source data file. Source data are provided with this paper.

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

## Acknowledgements

This research was supported by the National Institute of General Medical Science R35GM128562 (B.D.F.) and F32GM140718 (T.M.W.). A portion of this research was supported by NIH grant R24GM154185 and performed at the Pacific Northwest Center for Cryo-EM (PNCC) with assistance from Nancy Meyer and Marzia Miletto. Research reported in this publication was supported also by the National Cancer Institute Cancer Center Support Grant P30 CA168524. We would also like to acknowledge David Ingham at the University of Kansas Medical Center Electron Microscopy Research Laboratory for helpful discussions related to the manuscript.

## Author contributions

T.M.W., B.J.R., and B.D.F. conceptualized the experiments. T.M.W. and B.J.R. generated/purified the nucleosomes and Pol β proteins for biochemical assays and cryo-EM experiments. T.M.W., B.J.R., S.H.T., A.S.H., and J.J.S. performed and analyzed the biochemical experiments. T.M.W., B.J.R., Z.X., and N.J.S. performed cryo-EM sample preparation and validation. T.M.W., B.J.R., and B.D.F. processed and analyzed the cryo-EM datasets. T.M.W., B.J.R., and B.D.F. performed model building and refinement. T.M.W., B.J.R., and B.D.F. wrote the manuscript with input from Z.X. and N.J.S.

## Competing interests

The authors declare no competing interests.
