## [Transparent Peer Review file · Nature Communications]

Structural basis of gap-filling DNA synthesis in the nucleosome by DNA Polymerase β

Corresponding Author: Dr Bret Freudenthal

Version 0:

Reviewer comments:

Reviewer #1

(Remarks to the Author)

This manuscript by Weaver et al. explores how 1-nt gaps, a prevalent form of DNA damage, are processed by DNA polymerase beta ($\text{pol}\beta$) in the context of a nucleosome core particle (NCP). The authors utilize a series of NCP substrates with 1-nt gaps that vary in position. Cryo-EM structures show the engagement of $\text{pol}\beta$ with the 1-nt gaps. These structures captured the ability of $\text{pol}\beta$ to sculpt its substrate to better position the DNA into the active site for catalysis. This structural information is supported by single-turnover pre-steady state kinetics to investigate how the rate of nucleotide insertion varies. The authors report that the rate of nucleotide insertion is position dependent and that differences in affinity of the enzyme for the substrate do not cause these changes. These thorough studies provide new insights into the substrate processing by $\text{pol}\beta$ and will be a valuable contribution to the literature. The following revisions are recommended prior to publication:

- 1) Based on the structural insights gained, might the authors speculate why $\text{pol}\beta$ is unable to efficiently sculpt the DNA as needed at each position?
- 2) It's not clear why SHL-3.5 would be the site with the highest degree of unwrapping. Can the authors discuss this idea further?
- 3) While solvent accessibility of the 1-nt gaps was discussed, it was not described how solvent accessibility was determined.
- 4) Solvent accessibility for the three sites with different rotational positions (Gap-NCP-4.5+2, Gap-NCP-4.5+3, Gap-NCP-4.5+4) is discussed but the solvent accessibility for the other sites (Gap-NCP-1.5, Gap-NCP-2.5, Gap-NCP-3.5, Gap-NCP-4.5, Gap-NCP-5.5) is not discussed/described.
- 5) In the STK studies, 100% product is not observed and this result is interpreted as being due to the "decrease in the amount of productive $\text{pol}\beta$:Gap-NCP nucleotide insertion complex." Is it possible that a portion of the substrate is not chemically what you think and, therefore, not a substrate for $\text{pol}\beta$?
- 6) Lines 137-138 – A conclusion was made that the differences in $\text{pol}\beta$ nucleotide insertion rate observed at various positions are not a result of large differences in substrate affinity. But there are differences in K_d values. Can the authors discuss their interpretations further?
- 7) Line 239-241 – It appears as though the full length and lyase domain have different nonspecific affinities.
- 8) The dRP group is not present in these NCP. Can the authors discuss how this may influence recognition by $\text{pol}\beta$ in these experiments and/or interpretation of the results?
- 9) There have been prior biochemical studies of $\text{pol}\beta$ activity on nucleosomes. It would be helpful if the authors could include a comparison to those literature values including a discussion of the location of the 1-nt gap site.
- 10) EMSAs were performed in a different buffer than STK experiments. Are you able to directly compare this data knowing that enzyme activity on DNA is influenced by buffer conditions?

Minor Issues:

- 12) In some cases, figures are not correctly referenced in the text. For example, lines 96-97 discuss k_{obs} values and then reference figure 1c-d. Figure 1d does not describe k_{obs} .
- 13) Line 132 – it is unclear what is meant by "consistent with robust nucleosome binding".
- 14) Line 364 – typo, should be "complementary" rather than "complimentary"
- 15) Concentrations of $\text{pol}\beta$ are provided. Are those active or total concentrations?
- 16) The use of pink and purple to highlight the different translational and rotational positions in Figure 1 is very helpful. But

the same colors are then used to represent different polymerase domains in other figures. This could confuse the reader.

17) Figures 1e and 3a – This method of nucleobase labeling (dA-120, dG121, etc.) has not been explained.

18) Figure 3a – A more detailed explanation of the inserts in the caption would improve clarity.

Reviewer #2

(Remarks to the Author)

This study describes the cryo-EM structures and biochemical characterization of single nucleotide gap filling by DNA polymerase beta (Pol beta) within nucleosome core particles (NCP). Single nucleotide gaps were placed at various positions within the NCP with the goal of determining if positioning of these gaps with respect to the NCP alters the ability of Pol beta to fill the gaps and also to elucidate the underlying mechanism of gap filling. Based on their data the authors conclude that Pol beta uses a sculpting mechanism to fill the gap that involves a 90 degree bend and interaction of the lyase domain with the gap, then the polymerase domain, followed by catalysis. The distortions induced by the sculpting are suggested to unwrap the NCP. This mechanism appears to be similar at various positions of the gap within the NCP. However, the efficiency of gap filling decreases as the gap is less accessible to Pol beta due to positioning within the NCP and the authors propose that additional proteins are likely required to unwrap the NCP to provide access to Pol beta. Some of this work confirms previous biochemical work with Pol beta and other base excision repair proteins. However, the research presented here significantly extends results of previous studies through demonstration of a potential sculpting mechanism by Pol beta that is used for gap filling by this enzyme.

Although this study has a great deal of novelty and makes an impactful contribution to the field, it is felt that there are some concerns that the authors should consider addressing.

1. The authors employ the Widom sequence, which is known to be a highly stable as part of the NCP. This stability could impact the ability of Pol beta to fill in the gap within the context of the NCP. In contrast, sequences with less stability might yield different results and perhaps an alternative mechanism.
2. The rate of gap filling within the DNA sequence used by the authors is considered to be quite slow even in the absence of the NCP, limiting the dynamic range of measurement. It would be interesting to know if the authors have characterized catalysis by Pol beta within other sequence contexts (not the Widom) with and without the NCP.
3. It is unlikely that Pol beta is acting alone during gap filling. Other proteins within the single-strand break repair and/or base excision repair pathway are likely influencing gap filling and perhaps altering its mechanism with the NCP. Therefore, this work may not “establish the mechanism...” but rather be indicative of a particular mechanism.
4. Incorporation of different bases is known to be associated with altered kinetics. However, the authors do not seem to take this into account when comparing the positions of the gap with the NCP.
5. It is requested that the authors explain how they are certain that the reactions they conduct are actually being conducted under single turnover conditions. Were empirical measurements made?
6. The authors propose that the mechanism for the filling of gaps involves the binding of the lysase domain of Pol beta to the 5'-phosphate before engagement of the polymerase domain. Have experiments been conducted in the absence of the 5'-phosphate?
7. The apparent K_ds for DNA are similar for each of the constructs. It is suggested that the authors provide an explanation as to why they assume that Pol beta is engaging the gap in a similar manner in these experiments. The authors appear to have quantified disappearance of the free NCP rather than appearance of the complex in their EMSA experiments. It is recommended that the authors provide a rationale for this approach. Have the authors considered performing active site titration experiments to ensure that they are measuring a productive catalytic complex?

Reviewer #3

(Remarks to the Author)

This manuscript describes how Polymerase beta fixes gaps on a nucleosome, using a combination of in vitro approaches. Several high-quality cryoEM structures are complemented by well-controlled functional assays. The data are of high technical quality. It is shown that Polymerase β can bind to and act on nucleosomes with gaps, and that the efficiency is position dependent (rotational and translational). These experiments are well-executed and convincing, and the results are not surprising. The authors then proceed to determine cryoEM structures of the polymerase bound to the various gapped nucleosomes, after showing that they all bind with comparable affinities. The nucleosomes are well resolved, the polymerase less so, as is usually the case with these systems. However, the resolution is sufficient to see what is going on, in particular how the gapped DNA is lifted off and engaged with the polymerase, similar to what is observed with free DNA.

This requires the breaking of the histone-DNA contacts. Of note, it is not stated in the main text that all complexes were crosslinked. Overall, the manuscript provides a nice model for how Polbeta engages the nucleosome, with surprising insight into the degree of displacement of the DNA from the nucleosome. The figures are very clear, mostly, and the manuscript is well written. I only have a few minor quibbles.

- The nucleosomes look beautiful, but the authors should explain why they migrate differently in Fig. S1. Spell out what ND-NCP stands for in the figure legend.
- It is not spelled out what their assay actually is. E.g. it would help to describe what is actually being measured in fig. 1c in the main text; as it stands it is not even described in Sup Fig. 1 figure legend. I realize that this is a standard assay in the lab, but the readers won't be familiar with it and we shouldn't have to go to the methods section to learn what is actually being measured.
- There are no page numbers, but on page 5 (I think) here the verb is 'affect' 'To determine how the translational position of the 1-nt gap effects Pol nucleotide insertion'
- Supplementary Figure 2: what are the R2 for the fits? What binding model was used?
- Fig. 4d – not clear what the left-hand arrow in the left panel indicates – doesn't seem like a 17 Å movement to me.
- Page 11: 'and and' (two different occasions on this page)
- End of page 11, not sure whether the SHL-3.5 is the one that they are referring to as 'close to the dyad', please clarify language. As a note of interest, have the authors added gaps that really are close to the dyad?
- Discussion: I don't know what a 'global sculpting mechanism' is. looks pretty local to me. And why do they think it is 'interesting' that it the sculpting mechanism is more extensive than on free DNA? I would have expected that. Also please don't use the word 'significantly' in this context as there is no statistical analysis.
- Page 13, starting with 'Prior studies' – this sentence needs some grammatical help.
- The paragraph starting with 'the results of our kinetic and structural analysis..' (page 14) is overstated; the data actually doesn't suggest that, but rather it's a common-sense assumption even without the data presented here. All gaps that are studied here were fixed; and unless they identified gaps that emphatically are NOT fixed in vitro this statement has to be modified.
- We only learn in the methods section that the particles were crosslinked. This should be pointed out earlier, and mentioned in the discussion as a source for potential artifacts.

Signed - Karolin Luger

Version 1:

Reviewer comments:

Reviewer #1

(Remarks to the Author)

My questions and concerns have been sufficiently addressed in this revised manuscript.

Reviewer #2

(Remarks to the Author)

In general, the authors were responsive to the critiques. The additions to the manuscript provide additional clarity. Some concerns remain regarding the rate of pol beta catalysis even in non-nucleosomal DNA.

Reviewer #3

(Remarks to the Author)

I am satisfied with the revisions, all of my points have been addressed thoroughly and carefully. I appreciate the attention to detail. I am also gratified that the authors rebutted my criticism of their use of the word 'significant' (unless warranted by statistical analysis) with the following: 'We have made a significant effort to alleviate this confusion throughout the manuscript'. (:). At any rate, congratulations to all authors on a beautiful story and manuscript (signed K. Luger)

Reviewer #1 (Remarks to the Author):

This manuscript by Weaver et al. explores how 1-nt gaps, a prevalent form of DNA damage, are processed by DNA polymerase beta (pol β) in the context of a nucleosome core particle (NCP). The authors utilize a series of NCP substrates with 1-nt gaps that vary in position. Cryo-EM structures show the engagement of pol β with the 1-nt gaps. These structures captured the ability of pol β to sculpt its substrate to better position the DNA into the active site for catalysis. This structural information is supported by single-turnover pre-steady state kinetics to investigate how the rate of nucleotide insertion varies. The authors report that the rate of nucleotide insertion is position dependent and that differences in affinity of the enzyme for the substrate do not cause these changes. These thorough studies provide new insights into the substrate processing by pol β and will be a valuable contribution to the literature. The following revisions are recommended prior to publication:

Response: We appreciate the reviewer's overall enthusiasm for our work and the acknowledgement that it will be a valuable contribution to the field. We have addressed the comments/concerns of the reviewer in the point-by-point response below. All page and line numbers herein reference the location of the text within the revised manuscript. We have also highlighted regions of the revised manuscript with major textual changes in red.

1) Based on the structural insights gained, might the authors speculate why pol β is unable to efficiently sculpt the DNA as needed at each position?

Response: We do believe our data provides a plausible rationale for the inability of Pol β to sculpt the nucleosomal DNA and efficiently catalyze nucleotide insertion at different positions in the nucleosome. The position-dependent nucleotide insertion by Pol β seems to be dictated by which regions of the nucleosome can readily accommodate displacement of the nucleosomal DNA from the histone octamer, 90° bending of the nucleosomal DNA, and repositioning the 5'-phosphate and the primer terminal 3'-OH into the lyase and polymerase active sites, respectively. Translational and/or rotational positions in the nucleosome where any one of these three structural requirements is impeded will likely result in a reduction in Pol β nucleotide insertion.

This is a key aspect of the manuscript, which we feel was not conveyed sufficiently in the original manuscript. We have updated the discussion and point Reviewer 1 to the revised manuscript (page 14-15, lines 343-365), where an entire paragraph is now dedicated to better explaining our working model of the structural basis for position-dependent enzymatic activity of Pol β .

2) It's not clear why SHL-3.5 would be the site with the highest degree of unwrapping. Can the authors discuss this idea further?

Response: This question is intimately related to the reviewer's first question, and we point the reviewer to the updated discussion section, which clarifies our hypothesis for this phenomenon (page 14-15, lines 343-365). We briefly summarize below:

This observation at SHL-3.5 is likely due to the inability of the nucleosome to accommodate the localized 90° bend surrounding the Pol β binding site while simultaneously maintaining the canonical trajectory of the nucleosomal DNA around the histone octamer. This structural constraint at SHL-3.5 can be overcome through unwrapping of the nucleosomal DNA at the proximal entry/exit site. In contrast, the localized 90° bend surrounding the Pol β binding site can be readily accommodated at SHL-5.5 and SHL-4.5 without changing the canonical trajectory of

the remaining nucleosomal DNA around the histone octamer. We do appreciate that this is likely an oversimplification of the exact mechanism, but this is our working model based on the data currently available.

3) While solvent accessibility of the 1-nt gaps was discussed, it was not described how solvent accessibility was determined.

Response: Solvent-accessibility was inferred from multiple cryo-EM structures of nucleosomes determined with the exact Widom 601 sequence used in this manuscript (see Weaver et al., Nature Communications, 2022; Cordero, Mehta and Weaver et al., Nature Communications, 2024). To address the reviewer's concern, we have clarified this point in the manuscript. "Importantly, the translational position and rotational orientation of these 1-nt gaps were defined based on previously determined cryo-EM structures of NCPs with the same nucleosomal DNA sequence^{41,42}" (page 5, lines 89-91).

4) Solvent accessibility for the three sites with different rotational positions (Gap-NCP-4.5+2, Gap-NCP-4.5+3, Gap-NCP-4.5+4) is discussed but the solvent accessibility for the other sites (Gap-NCP-1.5, Gap-NCP-2.5, Gap-NCP-3.5, Gap-NCP-4.5, Gap-NCP-5.5) is not discussed/described.

Response: The solvent accessibility for the 1-nt gap positions with different translational positions was initially described during the introduction of the eight NCPs we generated with 1-nt gaps at different positions. See "Five of the NCPs contain a solvent-exposed 1-nt gap in a unique translational position (i.e., position relative to the nucleosome dyad) at SHL-5.5, SHL-4.5, SHL-3.5, SHL-2.5, and SHL-1.5" (page 5, lines 85-87).

5) In the STK studies, 100% product is not observed and this result is interpreted as being due to the "decrease in the amount of productive pol β :Gap-NCP nucleotide insertion complex." Is it possible that a portion of the substrate is not chemically what you think and, therefore, not a substrate for pol β ?

Response: We understand the reviewer's concern. We also considered this possibility, though we think it is highly unlikely. The most likely source of possible chemical heterogeneity in the oligonucleotides would be truncation products arising during the oligonucleotide synthesis. These truncation products would result in gap sizes greater than 1-nt after nucleosome reconstitution, which we agree could impact the analysis. However, the reduction in the amplitude of product formation for the different Gap-NCPs does not correlate with the length of either of the two oligonucleotides that are used to form the discontinuous strand containing the 1-nt gap (see Supplementary Table 4). In addition, a reduction in the amplitude of product formation has been previously observed for several DNA glycosylases (Tarantino et al., DNA Repair, 2018; Olman et al., ACS Chemical Biology), AP-endonuclease I (Weaver et al., Nature Communication, 2022) and Pol β (Rodriguez et al., JBC, 2016) when performing single-turnover kinetics experiments with recombinant nucleosomes. In these additional examples, the magnitude of the reduction in the amplitude of product formation is directly correlated to the translational and rotational position of the DNA damage in the nucleosome. Taken together, the simplest explanation for our observations is that the position of the 1-nt gap within the nucleosome directly impacts the amount of productive Pol β -Gap-NCP nucleotide insertion complex that is formed, rather than chemically heterogeneous oligonucleotides.

6) Lines 137-138 – A conclusion was made that the differences in pol β nucleotide insertion rate observed at various positions are not a result of large differences in

substrate affinity. But there are differences in Kd values. Can the authors discuss their interpretations further?

Response: We are happy to further clarify this interpretation. Our rationale for this interpretation is that the differences in apparent binding affinity for Pol β and the nucleosomal 1-nt gaps at different translational and rotational orientations are generally orders of magnitude smaller than the differences we found for k_{obs} . For example, the apparent binding affinity of Pol β for Gap-NCP-5.5 and Gap-NCP-4.5+2 are 51 ± 3 nM and 137 ± 11 nM, respectively. This is a modest ~ 2.6 x difference in apparent binding affinity between the two nucleosomal positions. In stark contrast, the k_{obs} for Pol β nucleotide insertion at Gap-NCP-5.5 and Gap-NCP-4.5+2 was $0.12 \pm 2.7 \times 10^{-3} \text{ s}^{-1}$ and $4.9 \times 10^{-4} \pm 5.1 \times 10^{-5} \text{ s}^{-1}$, respectively. This is an ~ 245 x difference in nucleotide insertion rate between the two nucleosomal positions. Though we use these two nucleosomal 1-nt gap positions as an example, this observation is true amongst all the 1-nt positions we tested in the nucleosome (see Supplementary Table 1). We believe these comparisons strongly suggest that the major driver of the reduction in Pol β nucleotide insertion rate (k_{obs}) at different 1-nt gap positions in the nucleosome is not the result of the subtle differences observed in apparent binding affinity.

To improve clarity, we have added an additional sentence stating, "*Though we did observe small differences in Pol β apparent binding affinity for the 1-nt gaps at each translational and rotational position in the nucleosome, these differences are orders of magnitude smaller than the reduction in k_{obs} (compare Fig. 1c,e and Supplementary Table 1)*" (page 7, lines 144-147).

7) Line 239-241 – It appears as though the full length and lyase domain have different nonspecific affinities.

Response: We appreciate the reviewer pointing out the ~ 2 x difference in non-specific binding for full-length Pol β and the Pol β lyase domain. We have re-written this section to point out this subtle difference in non-specific binding for full-length Pol β and the Pol β lyase domain (page 11, lines 251-260).

8) The dRP group is not present in these NCP. Can the authors discuss how this may influence recognition by pol β in these experiments and/or interpretation of the results?

Response: As noted by the reviewer, Pol β possess both 1-nt gap-filling activity and a 5'-dRP lyase activity. In non-nucleosomal DNA, Pol β can perform gap-filling and 5'-dRP lyase activity during a single binding event (Prasad et al., JBC, 2010; Kumar et al., PNAS, 2022), and kinetic studies indicate the gap filling likely occurs in the presence of the 5'-dRP moiety in non-nucleosomal DNA (Prasad et al., JBC, 1998; Prasad et al., JBC, 2010; Kumar et al., PNAS, 2022). Interestingly, the nucleosome has minimal impact on Pol β 5'-dRP lyase activity (Rodriguez et al., NAR, 2017), which is in stark contrast to the impaired 1-nt gap filling activity in the nucleosome (this manuscript and manuscript references 32-41). Given this experimental data, we hypothesize the 5'-dRP is not present during the 1-nt gap filling reaction in the nucleosome. In fact, the structure of the 5'-capture state intermediate (Fig. 4, state 1) may represent a structural state where 5'-dRP lyase activity could occur prior to the polymerase domain ever engaging the primer terminus of the nucleosomal DNA. However, it's impossible to know whether a similar 5'-capture state intermediate occurs when Pol β interacts with a 5'-dRP in the nucleosome. We have chosen not to speculate on this interesting topic in the manuscript discussion, though we anticipate future structural and kinetic studies will investigate the interplay between the gap-filling activity and 5'-dRP lyase activity of Pol β within the nucleosome.

- 9) There have been prior biochemical studies of pol β activity on nucleosomes. It would be helpful if the authors could include a comparison to those literature values including a discussion of the location of the 1-nt gap site.**

Response: We agree with the reviewer that prior biochemical studies have been performed with Pol β and nucleosomes with 1-nt gaps, which we referenced in our introduction (page 4, lines 72-74). However, many of these experiments (see manuscript references 32-41) were done under different buffer conditions, different kinetic regimes, and/or performed via multi-enzyme reconstitution assays (e.g. UDG, APE1, Pol β). These confounding factors make it difficult to directly compare these kinetic parameters with those obtained in our manuscript. We strongly feel that this type of detailed comparative analysis would be better suited for a review/perspective on Pol β function in the nucleosome, something we envision writing in the future, rather than in the discussion of the current manuscript. Though we previously noted these biochemical studies in our introduction (page 4, lines 72-74), we have also added an additional statement in the discussion highlighting that, “...these trends in enzymatic activity in the nucleosome are generally consistent with prior observations made for Pol β ²⁸⁻³⁷ and upstream BER/SSBR enzymes on defined nucleosome substrates^{53,55-57}” (page 14, lines 340-342).

- 10) EMSAs were performed in a different buffer than STK experiments. Are you able to directly compare this data knowing that enzyme activity on DNA is influenced by buffer conditions?**

Response: We understand the reviewer's concern regarding the slightly different buffers between the STK experiments and the EMSA experiments. The core components of the buffer (i.e. salt concentration, pH, reducing reagent, and crowding agent) are virtually identical between the experiments. Some of the auxiliary components of the buffers are different between the two sets of experiments (i.e. Mg₂Cl, dNTP, and glycerol). However, this unfortunately becomes unavoidable as the STK experiments require incoming dNTP and divalent cation metal (i.e. Mg²⁺) to initiate the nucleotide insertion reaction. In contrast, the EMSA experiments are performed in the absence of incoming dNTP and divalent cation metal (i.e. Mg²⁺) to avoid catalysis, as nucleotide insertion in these experiments would result in a Pol β apparent binding affinity for each Gap-NCP that reflects both 1-nt gap binding (substrate) and 5'-nick binding (nucleotide insertion product). Given that the core components are virtually identical, we feel it is appropriate to compare these data within the manuscript.

Minor Issues:

- 11) In some cases, figures are not correctly referenced in the text. For example, lines 96-97 discuss kobs values and then reference figure 1c-d. Figure 1d does not describe kobs.**

Response: We thank the reviewer for pointing out these errors. We have gone back through the entire manuscript to ensure all figures are correctly referenced.

- 12) Line 132 – it is unclear what is meant by “consistent with robust nucleosome binding”.**

Response: We have reworded this statement to “The $K_{d, app}$ of Pol β for the ND-NCP was 217 ± 14 nM, indicating Pol β can bind the nucleosome even in the absence of a 1-nt gap” to improve clarity (page 7, lines 139-140).

13) Line 364 – typo, should be “complementary” rather than “complimentary”

Response: We thank the reviewer for pointing out this grammatical error. The grammatical error has been fixed.

14) Concentrations of pol β are provided. Are those active or total concentrations?

Response: The concentrations of Pol β that were provided are total concentrations of enzyme. We have added an additional statement in the methods section describing how the concentrations of Pol β were determined and a statement informing the reader that total Pol β concentrations were used throughout the manuscript, rather than active Pol β concentration (page 17, lines 402-405).

15) The use of pink and purple to highlight the different translational and rotational positions in Figure 1 is very helpful. But the same colors are then used to represent different polymerase domains in other figures. This could confuse the reader.

Response: We respect the reviewer’s opinion. However, we disagree that the current color scheme will lead to confusion amongst future readers of our work, as the color scheme used within each of the figures and panels is clearly labeled.

16) Figures 1e and 3a – This method of nucleobase labeling (dA-120, dG121, etc.) has not been explained.

Response: We have defined the nomenclature used for nucleobase labeling in the figure legends for Fig. 2 (page 34, lines 841-849) and Fig. 3 (page 36, lines 857-859).

17) Figure 3a – A more detailed explanation of the inserts in the caption would improve clarity.

Response: We have added an additional sentence to the Fig. 3 figure legend describing the structural changes in the DNA induced by Pol β in the three insets to improve clarity (page 36, lines 860-863).

Reviewer #2 (Remarks to the Author):

This study describes the cryo-EM structures and biochemical characterization of single nucleotide gap filling by DNA polymerase beta (Pol beta) within nucleosome core particles (NCP). Single nucleotide gaps were placed at various positions within the NCP with the goal of determining if positioning of these gaps with respect to the NCP alters the ability of Pol beta to fill the gaps and also to elucidate the underlying mechanism of gap filling. Based on their data the authors conclude that Pol beta uses a sculpting mechanism to fill the gap that involves a 90 degree bend and interaction of the lyase domain with the gap, then the polymerase domain, followed by catalysis. The distortions induced by the sculpting are suggested to unwrap the NCP. This mechanism appears to be similar at various positions of the gap within the NCP. However, the efficiency of gap filling decreases as the gap is less accessible to Pol beta due to positioning within the NCP and the authors propose that additional proteins are likely required to unwrap the NCP to provide access to Pol beta. Some of this work confirms previous biochemical work with Pol beta and other base excision repair proteins. However, the research presented here significantly extends results of previous studies through demonstration of a potential sculpting mechanism by Pol beta that is used for gap filling by this enzyme.

Although this study has a great deal of novelty and makes an impactful contribution to the field, it is felt that there are some concerns that the authors should consider addressing.

Response: We thank the reviewer for their kind words on the novelty of our work and the potential impact of this work on the field. We have addressed the comments/concerns of the reviewer in the point-by-point response below. All page and line numbers herein reference the location of the text within the revised manuscript. We have also highlighted regions of the revised manuscript with major textual changes in red.

1. The authors employ the Widom sequence, which is known to be a highly stable as part of the NCP. This stability could impact the ability of Pol beta to fill in the gap within the context of the NCP. In contrast, sequences with less stability might yield different results and perhaps an alternative mechanism.

Response: We chose to utilize the Widom 601 sequence in the experiments throughout the manuscript because of its strong positioning ability. The rationale for this was two-fold. First, using the Widom 601 sequence allowed us to precisely position the 1-nt gap in the nucleosome in terms of translational position and rotational orientation. This allowed us to probe questions using our kinetic analysis that would generally be intractable with genomic sequences that lack the ability to strongly position the 1-nt gap. Second, the general stability of nucleosomes generated using the Widom 601 sequence becomes quite advantageous when trying to generate NCP-enzyme complexes for single particle analysis. This explains why so few structures of NCP-protein complexes exist that contain non-Widom 601 sequences.

Of course, we agree with the reviewer that utilizing alternative sequences could impact the findings, though this is likely true for the kinetic analysis more so than the structural analysis. For example, alternative sequences could possibly dampen the reduction in Pol β nucleotide insertion rate for 1-nt gaps near the nucleosome dyad or histone-occluded 1-nt gaps, as these sequences would have a higher propensity to fluctuate in rotational orientation and in some cases translational position. In contrast, we do not envision this would impact the structural mechanism

(i.e. global DNA sculpting by Pol β) as this is the same mechanism used to engage 1-nt gaps in non-nucleosomal DNA (Fig. 2g).

We absolutely agree that a systematic characterization of Pol β nucleotide incorporation at multiple positions within a non-Widom 601 DNA sequence would be an interesting and useful comparison. One of the major reasons we did not expand our kinetic analysis to alternative sequences is the difficulty in monitoring single nucleotide insertion on long oligonucleotides like the ones used to reconstitute a nucleosome (see Supplementary Table 4). Because of this, the kinetic analysis in Fig. 1 alone took over a year to finish, and mirroring those experiments in a non-Widom 601 sequence would likely take an additional year or more to complete. We strongly feel this type of systematic kinetic analysis on non-Widom 601 sequences is best suited for a future dedicated manuscript. Nevertheless, we have added a sentence in the discussion highlighting that genomic sequences that are less stable in positioning the histone octamer compared to the Widom 601 sequence may alter the results from our kinetic analysis (page 14, lines 335-337).

2. The rate of gap filling within the DNA sequence used by the authors is considered to be quite slow even in the absence of the NCP, limiting the dynamic range of measurement. It would be interesting to know if the authors have characterized catalysis by Pol beta within other sequence contexts (not the Widom) with and without the NCP.

Response: We thank the reviewer for this insightful comment. We do note that this comment is very much related to the comment #1 above, and many aspects of that response are applicable here as well.

The Pol β nucleotide insertion rate (k_{obs}) we determined in the non-nucleosomal Widom 601 Gap-DNA (147 bp) was $0.30 \pm 0.02 \text{ s}^{-1}$, which can be considered quite slow depending on what prior values are used for comparison. For example, this compares reasonably well with The Pol β nucleotide insertion rate (k_{obs}) of 0.12 s^{-1} previously reported for a 31 bp Gap-DNA (Liu et al., JBC, 2005). However, this value is roughly 10x slower than the maximum rate constant for nucleotide insertion (k_{pol}) determined in other studies (see Werneberg et al., Biochemistry, 1996; Varela and Freudenthal, Biochemistry, 2021; Kumar et al., PNAS, 2022, for several examples). While there are many possibilities for these difference in Pol β nucleotide insertion rate in non-nucleosomal Gap-DNA, a likely source of these differences is the length of the oligonucleotides and/or subtle differences in reaction conditions. The differences in Pol β nucleotide insertion rate (k_{obs}) and the maximum rate constant for nucleotide insertion (k_{pol}) in non-nucleosomal Gap-DNA across the literature were precisely why we performed the control experiment ourselves using the exact DNA sequence we used to reconstitute the nucleosomes.

For the sake of the reviewer's curiosity, we did find an example of Pol β steady state enzyme kinetics that were performed on a nucleosome containing the *Xenopus borealis* 5S rDNA sequence, a weaker positioning sequence than the Widom 601 sequence, during our initial literature review (Rodriguez et al., JBC, 2016). However, we could not find the control experiment performed using the 5S rDNA sequence in the absence of the histone octamer in that manuscript, and the use of steady state enzyme kinetics make it difficult for comparison to the k_{obs} we determined here from single-turnover kinetics experiments. In the future, it will be interesting to test whether alternative non-601 DNA sequences change the dynamic range of measurement for Pol β nucleotide insertion and/or the position-dependent Pol β nucleotide insertion rate in the nucleosome.

3. It is unlikely that Pol beta is acting alone during gap filling. Other proteins within the single-strand break repair and/or base excision repair pathway are likely influencing gap filling and perhaps altering its mechanism with the NCP. Therefore, this work may not “establish the mechanism...” but rather be indicative of a particular mechanism.

Response: We agree with the reviewer that additional SSBR/BER proteins likely influence this mechanism, which is why we dedicated a section of the discussion to highlight how additional factors almost assuredly impact the ability of Pol β to perform gap-filling DNA synthesis in the nucleosome. We have also softened the text in the abstract (page 2, lines 35-37).

4. Incorporation of different bases is known to be associated with altered kinetics. However, the authors do not seem to take this into account when comparing the positions of the gap with the NCP.

Response: We thank the reviewer for bringing up this excellent point. Indeed, the Pol β nucleotide insertion rate in non-nucleosomal DNA is different depending on the identity of the templating base and the incoming dNTP (Beard et al., JBC, 2002; Beard et al., JBC, 2004). We did heavily consider this when initially designing our experiments. However, we decided to keep the Widom 601 DNA sequence intact and not change the sequence to match the templating base and incoming dNTP at all positions tested. This was a calculated decision based out of concern that changing the nucleotide sequence of the Widom 601 DNA would possibly induce changes in nucleosome structure, stability, and/or dynamics that would complicate interpreting the Pol β nucleotide insertion rates across the different Gap-NCPs. This is a double-edge sword, as the Pol β nucleotide insertion rate can also vary depending on the identity of the templating base and incoming dNTP. We have added a sentence in the discussion highlighting this important point and stating that additional experiments will be needed to address this in the future (page 14, lines 337-339).

5. It is requested that the authors explain how they are certain that the reactions they conduct are actually being conducted under single turnover conditions. Were empirical measurements made?

Response: The concentration of Pol β (1 μ M) used in the single-turnover kinetics experiments was chosen following completion of our EMSA experiments to ensure the experiments would be performed with the concentration of Pol β well above the apparent binding affinities for the Gap-NCPs (see Fig. 1e and Supplementary Table 1). While we would have preferred to verify the reactions were truly under single-turnover condition by performing additional experiments showing that the k_{obs} was unchanged at higher Pol β concentrations, we were severely limited in the quantity of Gap-NCP we had to do each of these experiments. To provide some context to the reviewer, the kinetic analysis in Fig. 1 alone took over a year to finish. Performing those additional experiments at multiple concentrations of Pol β for each different Gap-NCP would have likely taken us an additional year or more to complete.

6. The authors propose that the mechanism for the filling of gaps involves the binding of the lysase domain of Pol beta to the 5'-phosphate before engagement of the polymerase domain. Have experiments been conducted in the absence of the 5'-phosphate?

Response: We did not perform additional experiments with nucleosomes containing a 1-nt gap in the absence of the 5'-phosphate. Instead, we chose to do the reciprocal experiments to determine that the truncated form of Pol β containing only the lyase domain is sufficient to drive high affinity nucleosome binding and 1-nt gap specificity (Fig. 4e and Supplementary Fig. 10). Prior work

identified that the ability of Pol β to bind 1-nt gaps and perform gap-filling DNA synthesis in non-nucleosomal DNA is sensitive to the presence of the 5'-phosphate (Singhal and Wilson, JBC, 1993; Prasad et al., JBC, 1994; Huang et al., JBC, 2018). We expect this would also be true for nucleosomal DNA.

7. The apparent Kds for DNA are similar for each of the constructs. It is suggested that the authors provide an explanation as to why they assume that Pol beta is engaging the gap in a similar manner in these experiments. The authors appear to have quantified disappearance of the free NCP rather than appearance of the complex in their EMSA experiments. It is recommended that the authors provide a rationale for this approach. Have the authors considered performing active site titration experiments to ensure that they are measuring a productive catalytic complex?

Response: We appreciate the reviewer bringing these concerns to our attention. We have chosen to address these three concerns in the order they were brought up by the reviewer.

Unfortunately, we found it difficult to determine exactly what the reviewer is referring to regarding the first point, "*It is suggested that the authors provide an explanation as to why they assume that Pol beta is engaging the gap in a similar manner in these experiments.*" We do want to highlight that we made multiple changes in the sections of the manuscript describing the EMSA experiments for full-length Pol β (page 7, lines 136-149) and the EMSA experiments comparing full-length Pol β to the Pol β lyase domain (page 11, lines 251-260) to address minor concerns from Reviewer 1 and Reviewer 3. We believe these changes address the reviewer's original concern.

As noted by the reviewer, we determined the apparent binding affinity ($K_{d, app}$) of Pol β for a non-damaged NCP (ND-NCP, control experiment) and the eight different Gap-NCPs. For clarity, we would like to point out that these experiments were performed with NCPs, not DNA. The EMSA experiments were quantified via the disappearance of the free ND-NCP or Gap-NCP band, rather than the appearance of the Pol β -ND-NCP or Pol β -Gap-NCP complex band, and the quantified data fit to a one-site binding model accounting for ligand depletion to determine the apparent binding affinity ($K_{d, app}$). In these experiments, Pol β generates two types of lower mobility species upon nucleosome binding: (1) a smeared lower mobility species, and (2) a defined lower mobility species. While it would be reasonable to assume that the more defined lower mobility species in these experiments represents "specific" 1-nt gap binding, the same defined lower mobility species is observed in the absence of the 1-nt gap (Supplementary Fig. 2a). Because this band clearly does not represent only "specific" 1-nt gap binding, we chose to quantify the experiments via the disappearance of the NCP rather than the appearance of the Pol β -NCP or Pol β -Gap-NCP complex band, which allowed us to account for all Pol β -bound species in the experiment (both specific and non-specific). We also could have quantified by summing the appearance of all lower mobility species, but quantifying the intensity of the smears above background was problematic, so we chose to quantify the experiments as described above. We do understand that the EMSA experiments have technical limitations, particularly with enzymes that bind nucleic acids in both a specific and non-specific manner, and we are currently working on developing single-molecule approaches to directly monitor specific lesion recognition by BER enzymes in the nucleosome to overcome these limitations in the future.

We did consider performing active site titration experiments for Pol β and the Gap-NCPs while working up this manuscript. However, we are not sure these experiments will get after the question of "specific" Gap-NCP binding that we believe the reviewer is suggesting these experiments will answer. We believe this confusion could stem from an assumption that any Pol β engaged with a

1-nt gap in the nucleosome must be able to catalyze nucleotide insertion. Our structural snapshots of Pol β engaged with the nucleosomal 1-nt gap in conformations that are incompatible with nucleotide insertion (see Fig. 4, State 1 and State 2) indicate that Pol β can be engaged with the 1-nt gap but not in a conformation poised for nucleotide insertion. For example, the lyase domain can be engaged with the 5'-phosphate (i.e. bound to the 1-nt gap) while the polymerase domain remains disengaged from the primer terminus (Fig. 4, State 1). Because of this, we cannot assume that an indirect measurement of binding affinity from an active site titration (varying Gap-NCP concentration) would be any more representative of the true Pol β binding affinity for nucleosomal 1-nt gaps than the experiments we have already performed. Furthermore, generating the nucleosomal substrates again to perform these additional active site titration experiments is expensive and time-consuming with no guarantee that the results will provide additional information on 1-nt gap specificity in the nucleosome.

Reviewer #3 (Remarks to the Author):

This manuscript describes how Polymerase beta fixes gaps on a nucleosome, using a combination of in vitro approaches. Several high-quality cryoEM structures are complemented by well-controlled functional assays. The data are of high technical quality. It is shown that Polymerase β can bind to and act on nucleosomes with gaps, and that the efficiency is position dependent (rotational and translational). These experiments are well-executed and convincing, and the results are not surprising. The authors then proceed to determine cryoEM structures of the polymerase bound to the various gapped nucleosomes, after showing that they all bind with comparable affinities. The nucleosomes are well resolved, the polymerase less so, as is usually the case with these systems. However, the resolution is sufficient to see what is going on, in particular how the gapped DNA is lifted off and engaged with the polymerase, similar to what is observed with free DNA. This requires the breaking of the histone-DNA contacts. Of note, it is not stated in the main text that all complexes were crosslinked. Overall, the manuscript provides a nice model for how Pol beta engages the nucleosome, with surprising insight into the degree of displacement of the DNA from the nucleosome. The figures are very clear, mostly, and the manuscript is well written. I only have a few minor quibbles.

Response: We thank the reviewer for their overall enthusiasm for our work. We have addressed the comments/concerns of the reviewer in the point-by-point response below. All page and line numbers herein reference the location of the text within the revised manuscript. We have also highlighted regions of the revised manuscript with major textual changes in red.

1) The nucleosomes look beautiful, but the authors should explain why they migrate differently in Fig. S1. Spell out what ND-NCP stands for in the figure legend.

Response: We appreciate the reviewer pointing out the subtle differences in migration between some of the nucleosome samples. It is important to note that these samples were generated and run on native PAGE gels at different times over the course of a ~6-month period. The subtle differences in migration between the nucleosome samples are most likely due to small differences in electrophoresis time (see Source Data file SFig. 1a) or possibly small differences in electrophoresis reagents, which were generally made and used over the course of a week. We have added a sentence to the figure legend for Supplementary Fig. 1 legend to point out these subtle differences. We have also corrected the Supplementary Fig. 1 legend to explain that ND-NCP is an abbreviation for non-damaged NCP.

2) It is not spelled out what their assay actually is. E.g. it would help to describe what is actually being measured in fig. 1c in the main text; as it stands it is not even described in Sup Fig. 1 figure legend. I realize that this is a standard assay in the lab, but the readers won't be familiar with it and we shouldn't have to go to the methods section to learn what is actually being measured.

Response: We thank the reviewer for pointing out the lack of experimental description for the single-turnover kinetic experiments in the main text. We have added a couple of additional sentences to this paragraph to better describe what is being measured in the experiments (i.e. conversion of substrate to product) and what both calculated kinetic parameters (k_{obs} and amplitude of product formation) tell us about the reaction (page 5, lines 97-102).

3) There are no page numbers, but on page 5 (I think) here the verb is ‘affect’ ‘To determine how the translational position of the 1-nt gap effects Pol β nucleotide insertion’

Response: We thank the reviewer for pointing out this grammatical error. The grammatical error has been fixed.

4) Supplementary Figure 2: what are the R² for the fits? What binding model was used?

Response: The EMSA experiments were fit to a one site binding model accounting for ligand depletion. We have added a sentence to Supplementary Fig. 2 legend to inform the readers that the EMSAs were fit to this binding model and pointed the reader to the methods section where additional information is available. In addition, we have added the individual R² for the EMSA fits in Supplementary Fig. 2 (shown in the inset for each plot). For clarity, we have also done the same for the single-turnover kinetics experiments in Supplementary Fig. 1b and the associated figure legend, as well as the EMSA experiments with the Pol β lyase domain in Supplementary Fig. 10 and the associated figure legend.

5) Fig. 4d – not clear what the left-hand arrow in the left panel indicates – doesn’t seem like a 17 Å movement to me.

Response: We agree that the left-hand arrow in this figure panel was difficult to interpret. We initially had trouble finding an orientation that could readily show the movements of the 5'-phosphate, the 3'-OH, and the primer-terminal DNA simultaneously across all three structural states. We have added another panel to Fig. 4d with an alternative view that better highlights the 17 Å movement of the 5'-phosphate during initial 5'-phosphate capture by the lyase domain (i.e. Transition 1).

6) Page 11: ‘and and’ (two different occasions on this page)

Response: We appreciate the reviewer pointing out these grammatical errors. The grammatical errors have been fixed.

7) End of page 11, not sure whether the SHL-3.5 is the one that they are referring to as ‘close to the dyad’, please clarify language. As a note of interest, have the authors added gaps that really are close to the dyad?

Response: We apologize for the lack of clarity in this statement, which we have re-worded to improve clarity. *“...we observed stark differences in the displacement of the nucleosomal DNA by Pol β when the 1-nt gap was moved away from the nucleosome entry/exit site and towards the nucleosome dyad (i.e. from SHL-5.5/SHL-4.5 to SHL-3.5) (page 12, lines 281-283).*

We also appreciate the reviewer’s interest in 1-nt gaps positioned in closer proximity to the nucleosome dyad. We have added 1-nt gaps at positions closer to the nucleosome dyad than SHL-3.5. This includes 1-nt gaps at SHL-2.5 and SHL-1.5. We performed both EMSA experiments and single-turnover kinetic experiments for Pol β at these positions (see Fig. 1, Supplementary Fig. 1, and Supplementary Fig 2). If the reviewer is asking whether we made attempts to determine structures of Pol β bound to these 1-nt gap positions, we did indeed make single

attempts at determining cryo-EM structures of Pol β bound to Gap-NCP-2.5 and Gap-NCP-1.5. However, we were unsuccessful in obtaining reconstructions of Pol β bound to either Gap-NCP-2.5 or Gap-NCP-1.5. Due to time- and cost-limitations, we did not make additional attempts to optimize and determine these structures. We anticipate making additional attempts to determine these structures as well as other 1-nt gap positions in the future to further understand how DNA repair polymerases catalyze nucleotide insertion in the nucleosome.

8) Discussion: I don't know what a 'global sculpting mechanism' is. looks pretty local to me. And why do they think it is 'interesting' that it the sculpting mechanism is more extensive than on free DNA? I would have expected that. Also please don't use the word 'significantly' in this context as there is no statistical analysis.

We apologize that this section of the discussion was written in a way that confused the reviewer. We have made a significant effort to alleviate this confusion throughout the manuscript, predominately in the discussion section. Some of this confusion likely stemmed from the fact that we did not properly define what we mean by "DNA sculpting." We refer to DNA sculpting as the combination of structural changes in the nucleosomal DNA that result from 1-nt gap recognition by Pol β (pages 8-9, lines 182-198), similar to how an artist would sculpt clay. We understand that this may seem like a vague descriptor, but we feel it is a concept that can encompass the multiple different structural changes that Pol β , and other BER enzymes, generate when engaging DNA damage. We also strongly feel it also makes our structural findings more accessible to non-structural biologists.

The DNA sculpting mechanism used by Pol β in nucleosomal DNA is not more extensive than non-nucleosomal DNA, they are quite similar (see Fig. 2g). We believe the statement the reviewer is referring to was actually comparing DNA sculpting by Pol β and other upstream BER enzymes (page 13, lines 304-314). This statement was meant to imply that the "global" DNA sculpting mechanism used by Pol β , which impacts ~35 bp of nucleosomal DNA, is more extensive than the "localized" DNA sculpting mechanism used by upstream BER enzymes, which impact ~10 bp of nucleosomal DNA. This finding is interesting because it provides a rationale for why upstream BER enzymes (i.e. DNA glycosylases and APE1) are generally better at processing their cognate lesions in the nucleosome than downstream BER enzymes (i.e. Pol β). We updated that portion of the discussion to further clarify this point (page 13, lines 304-314). We also removed the word significantly from this section of the discussion.

9) Page 13, starting with 'Prior studies' – this sentence needs some grammatical help.

Response: We thank the reviewer for pointing out the grammatical issues and the lack of clarity in this opening sentence. We have adjusted the opening sentence in this paragraph to fix the grammatical issues and improve clarity for the reader (page 13, lines 315-319).

10) The paragraph starting with 'the results of our kinetic and structural analysis..' (page 14) is overstated; the data actually doesn't suggest that, but rather it's a common-sense assumption even without the data presented here. All gaps that are studied here were fixed; and unless they identified gaps that emphatically are NOT fixed in vitro this statement has to be modified.

Response: We would like to provide a brief rationale for stating "*The results of our kinetic and structural analysis strongly suggest that additional cellular factors are required to accelerate the*

ability of Pol β to process 1-nt gaps in chromatin, particularly at inaccessible locations in the nucleosome.”

Our kinetic analysis does show that certain less accessible regions of the nucleosomal DNA are refractory towards nucleotide insertion. For example, when the 1-nt gap was positioned in proximity to the nucleosome dyad (i.e. SHL-1.5), we detected >10% Pol β nucleotide insertion after 12 hours, which was not sufficient to even determine a nucleotide insertion rate. When the 1-nt gap was positioned in certain histone-occluded rotational orientations (i.e. SHL-4.5,+3 and SHL-4.5,+4) we only detected ~25% Pol β nucleotide insertion after 12 hours, whereas the remaining ~75% of the 1-nt gaps at these two positions are not repaired. This alone indicates that not all the 1-nt gaps are effectively processed by Pol β , which we envision would provide an opportunity for additional factors to help accelerate the processing of 1-nt gaps in the nucleosome.

Despite this clarification, we do agree with the reviewer that several statements in that paragraph can be toned-down and additional text added for clarity regarding our rationale for discussing how additional factor may accelerate 1-nt gap processing in the nucleosome. We have made multiple changes throughout the paragraph in an attempt to address the reviewer’s concerns (page 15-16, lines 366-379).

11) We only learn in the methods section that the particles were crosslinked. This should be pointed out earlier, and mentioned in the discussion as a source for potential artifacts.

We have added a statement to the introductory paragraph for the pre-catalytic Pol β -Gap-NCP-4.5 complex (page 7, lines 153-154) and the introductory paragraph for the Pol β -Gap-NCP-5.5 and Pol β -Gap-NCP-3.5 complexes (page 12, line 268) highlighting the complexes were stabilized using glutaraldehyde crosslinking.

The most likely source of artifacts arising from stabilization using glutaraldehyde crosslinking would be the structural snapshots we observed that show the sequential binding of the lyase and polymerase domains during 1-nt gap recognition in the nucleosome. We have added a sentence in the discussion highlighting this possibility, though we also explain why we think this is unlikely to be the case (page 14, lines 324-327).

Reviewer #1 (Remarks to the Author):

My questions and concerns have been sufficiently addressed in this revised manuscript.

Response: We thank the reviewer for the positive assessment of our manuscript.

Reviewer #2 (Remarks to the Author):

In general, the authors were responsive to the critiques. The additions to the manuscript provide additional clarity. Some concerns remain regarding the rate of pol beta catalysis even in non-nucleosomal DNA.

Response: We thank the reviewer for acknowledging our changes to the manuscript provided additional clarity for readers. To further alleviate the reviewer's concern, we have added an additional statement in the results section of the manuscript highlighting that the Pol β nucleotide insertion rate (k_{obs}) for the 147 bp Gap-DNA control is similar to the k_{obs} reported for a shorter 31 bp Gap-DNA substrate, but slightly slower than the maximum rate constant for Pol β nucleotide insertion (k_{pol}) found in the literature.

Reviewer #3 (Remarks to the Author):

I am satisfied with the revisions, all of my points have been addressed thoroughly and carefully. I appreciate the attention to detail. I am also gratified that the authors rebutted my criticism of their use of the word 'significant' (unless warranted by statistical analysis) with the following: 'We have made a significant effort to alleviate this confusion throughout the manuscript'. (;)). At any rate, congratulations to all authors on a beautiful story and manuscript (signed K. Luger)

Response: We thank the reviewer for the kind words regarding our manuscript. We also appreciate that the reviewer had a laugh at our unintentional blunder in the rebuttal.